# Learning Curves for Deep Neural Networks: A Field Theory Perspective

## Abstract

A series of recent works established a rigorous correspondence between very wide deep neural networks (DNNs), trained in a particular manner, and noiseless Bayesian Inference with a certain Gaussian Process (GP) known as the Neural Tangent Kernel (NTK). Here we leverage this correspondence to provide explicit analytical predictions for the learning curves of DNNs on any fixed target function. Focusing on datasets whose input measure is uniform on a hypersphere, our predictions match experimental curves very well and reveal a strong implicit bias towards functions which are low order polynomials of the input. Two novel analytical tools underlay our approach. First a renormalization-group approach is used to show that noiseless GP inference using NTK, which lacks a good analytical handle, can be well approximated by noisy GP inference on a related kernel we call the renormalized NTK. Second a field theory reformulation of the problem allows a controlled perturbative expansion of the test error in the inverse dataset size. We believe our approach lays the foundations for a more precise quantitative understanding of DNNs.

## 1 Introduction

Several pleasant features underlay the success of deep learning: The scarcity of bad minima encountered in their optimization (Draxler et al., 2018; Choromanska et al., 2014), their ability to generalize well despite being heavily over-parameterized (Neyshabur et al., 2018; 2014) and expressive (Zhang et al., 2016), and their ability to generate internal representations which generalize across different domains and tasks (Yosinski et al., 2014; Sermanet et al., 2013).

Due to the complexity of DNNs our current understanding of these features is still largely empirical. Notwithstanding, progress has been made recently in the highly over-parametrized regime (Daniely et al., 2016; Jacot et al., 2018) due to the fact that the networks' parameters, in all non-linear layers, change in a minor yet important manner during training. This facilitated the derivation of various bounds (Allen-Zhu et al., 2018; Cao & Gu, 2019b;a) on generalization and, more relevant for this work, the following correspondence with GPs: Considering finite-depth DNNs which are much wider than the dataset-size, trained with MSE loss, no weight decay, and at vanishing learning rate (the NTK-regime) one finds that the initialization-averaged predictions are the same as those of Gaussian Processes Regression (GPR) with a kernel known as the NTK. Several subsequent works corroborated these results empirically (Lee et al., 2018; Lee et al., 2019; Arora et al., 2019) and extended them (Arora et al., 2019). For fully-connected DNNs, the NTK-regime (and GPs associated with DNNs in general (Lee et al., 2018; Novak et al., 2018)) seems to faithfully capture the generalization power of DNNs trained with MSE loss (Lee et al., 2019).

One of the most detailed objects quantifying generalization are learning-curves: graphs of how the test error diminishes with the number of datapoints ($N$). There are currently no analytical predictions or bounds we are aware of for DNN learning-curves which are tight even just in terms of their scaling with $N$, let alone tight in an absolute sense. In contrast, the theory of GPR is rich with analytical tools which have yielded in the past high accuracy predictions for learning curves. One of the most transparent ones is the equivalence kernel (EK) (Rasmussen & Williams, 2005) which will be introduced in section 3. In short, EK gives an intuitive functional prediction of the expected GPR predictor of a fixed target function when averaged over datasets of size $N$, and consequently of the learning curves.

Clearly such a detailed understanding of generalization in DNNs is desirable. However, several technical issues prohibit the application of the EK and related results (Rasmussen & Williams, 2005; Malzahn & Opper, 2001) to DNNs trained in the NTK-regime. First, the NTK-regime corresponds to noiseless GPR where the DNN and corresponding GP both fit the training dataset exactly, whereas EK assumes Gaussian measurement noise on the training labels. In the case of noiseless GPR, EK and related approximations break down and predict perfect generalization with very small datasets. Second, the eigenfunctions (features) and eigenvalues of the NTK are needed so that the EK can be interpreted. Third, as EK results can be misleading, it is important to estimate the validity range of these approaches and, in a related manner, derive sub-leading corrections.

In this work we make the following contributions:

I We establish that noiseless GP inference using NTK can be well approximated by noisy GP inference on a related set of kernels we dub the renormalized NTKs.

II We obtain closed expression for the leading and sub-leading asymptotics of learning curves for any fixed target function (fixed-teacher learning curves) using an extension of the field-theory formalism of Malzahn & Opper (2001) for GPR.

III For uniform datasets, where the input is distributed uniformly on a hypersphere, these expression simplify considerably and, together with the features and eigenvalues we obtain for the renormalized NTKs, lead to clear relations between deep fully-connected networks and polynomial regression.

IV Again for uniform datasets, we provide analytical predictions for learning curves along with estimates for their range of validity. We believe our learning curves estimates stand-out in terms of accuracy as they get to within 3% of experimental generalization values. We emphasize that our predictions can be applied to fully-connected DNNs of any depth, trained in the NTK regime, and that our approach provides a systematic way of making further accuracy improvements.

## 2 PRIOR WORKS

Learning curves for GPs have been analyzed using a variety of techniques (see Rasmussen & Williams (2005) for a review) most of which focus on a GP-teacher averaged case where the target/teacher is drawn from the same GP used for inference (matched priors) and is furthermore averaged over. Fixed-teacher or fixed-target learning curves have been analyzed using a similar grand-canonical/Poisson-averaged approach (Malzahn & Opper, 2001) as our, however, the treatment of the resulting partition function was variational whereas we take a perturbation-theory approach. In addition previous cited results for MSE-loss breakdown in the noiseless limit (Malzahn & Opper, 2001). To the best of our knowledge, noiseless GPs learning-curves have been analyzed analytically only in the teacher-averaged case and in the following settings: For matched priors, exact results are known for one dimensional data (Williams & Vivarelli, 2000; Rasmussen & Williams, 2005) and two dimensional data with some limitations of how one samples the inputs (in the context of optimal design) (Ritter, 2007; 1996). In addition Micchelli & Wahba (1979) derived a lower bound on generalization. For noiseless inference with partially mismatched-priors (matching features, mismatching eigenvalues) and at large input dimension the teacher and dataset averaging involved in obtained learning curves has been performed analytically and the resulting matrix traces analyzed numerically Sollich (2001). Notably none of these cited results apply in any straightforward manner in the NTK-regime.

Considering kernel eigenvalues, explicit expression for the features and eigenvalues of dot-product kernels ($K = K(x \cdot x')$) were given in Azevedo & Menegatto (2015). The fact that the $l$-th eigenvalue of such kernels scales as $d^{-l}$ ($d$ being the input dimension), which we used in our derivation of the bound, has been noticed in Sollich (2001). Kernels with a trimmed spectrum where the spectrum is trimmed after the first $r$'s leading eigenvalues, has previously been suggested as a way of reducing the computational cost of GP inference (Ferrari-Trecate et al., 1998). In contrast we trim the Taylor expansion of the kernel function rather than the spectrum (which has a very different effect) and show that an effective observation noise compensates for our trimming/renormalization procedure.

Several interesting recent works give bounds on generalization (Allen-Zhu et al., 2018; Cao & Gu, 2019b;a) which show $O(1/\sqrt{N})$ asymptotic decay of the learning-curve (at best). In contrast our predictions are typically well below this bound.

## 3 FIELD THEORY FORMULATION OF GP LEARNING-CURVES

### 3.1 GPR DEFINITION AND PROBLEM STATEMENT

We begin with standard definitions of GPs and Bayesian Inference on GPs. A GP is defined as a stochastic process of which any finite subset of random variables follow a multivariate normal distribution. In a similar fashion to multivariate normal variables, GPs are also determined by their first and second moments. The first is typically taken to be zero, and second is known as the covariance function or the kernel $K_{xx'} = \mathrm{E}[f(x)f(x')]$, where $\mathrm{E}[\cdot]$ here denotes expectation under the GP distribution. The main appeal of GPs is that Bayesian Inference with GP priors is tractable. In GPR we use the posterior mean as the predictor $g^\star$, and it is given by:

$$g^\star(x_\star) = \sum_{n,m=1}^{N} K_{x_\star,x_n}[K(D) + \sigma^2 I]_{nm}^{-1} g_m \tag{1}$$

where $x_\star$ is a new datapoint, $g_m$ are the training targets, $x_n$ are the training data-points, $[K(D)]_{nm} = K_{x_n,x_m}$ is the covariance-matrix (the covariance-function projected on the training dataset $D$), and $\sigma^2$ is the variance of the assumed Gaussian noise of the labels, which also acts as a regulator of the prediction. Some intuition for this formula can be gained by verifying that in the noiseless case ($\sigma^2 = 0$) the prediction at some training point $x_\star = x_q$ coincides with that point's label $g^\star = g_q$.

The quantity of interest in this paper will be the dataset averaged generalization error which we define now. Throughout this paper we will assume that both train and test points are i.i.d. random variables drawn from a probability measure $\mu(x)$. With this in mind we define the generalization error of a prediction $g^\star$ or MSE loss as

$$\|g(x_\star) - g^\star(x_\star)\|^2 = \int d\mu_{x_\star} \, (g(x_\star) - g^\star(x_\star))^2 \tag{2}$$

Note that $g^\star$ is itself a function of $N$ draws from $\mu$ which make up the training set $D_N$. The dataset averaged generalization error is (2) averaged over the ensemble of all possible $N$ sized training sets. We denote this average as $\langle \cdot \rangle_{D_N}$, so for instance the dataset averaged generalization error would be $\langle \|g^\star - g\|^2 \rangle_{D_N}$. We see that in order to calculate learning-curves, one needs to average quantities like $g^\star$ and $g^{\star 2}$. These averages turn out to be difficult to handle analytically, so to facilitate their computation we adopt the approach of Malzahn & Opper (2001) and instead consider a related quantity given by the Poisson averaging of the former one

$$\langle ... \rangle_\eta = e^{-\eta} \sum_{n=0}^{\infty} \frac{\eta^n}{n!} \langle ... \rangle_{D_n} \tag{3}$$

where ... can be any quantity, in particular $g^\star$ and $g^{\star 2}$. Borrowing jargon from physics we refer to the original data ensemble as the canonical ensemble and to the above as the grand-canonical ensemble. Taking $\eta = N$, means we are essentially averaging over values of $N$ in an $\sqrt{N}$ vicinity of $N$. This means that as far as the leading asymptotic behavior is concerned, one can safely exchange $N$ and $\eta$ as the differences would be sub-leading. In App. A we compare learning curves as a function of $N$ and $\eta$ and show that they match very well.

Equation (1) determines the predictions, and therefore the learning-curves, but it is not very convenient for analytic exploration of the expected predictions. This fact is due to the (potentially very) large matrix inversion involved, and the additional averaging over $D_N$ required. Nonetheless there are some results for the expected prediction $g^\star$, the most famous of which is the equivalence kernel (EK) result for the prediction of a fixed target function as a function of the training set size $N$ (Rasmussen & Williams, 2005):

$$g^\star_{EK,N}(x) = \sum_n \frac{\lambda_n}{\lambda_n + \sigma^2/N} g_n \phi_n(x) \tag{4}$$

Where $\lambda_n$ and $\phi_n(x)$ here are the eigenvalues and eigenfunctions of the kernel w.r.t the input probability measure ($\mu$) and $g(x) = \sum_n g_n \phi_n(x)$ is the target. One notices immediately that the prediction breaks down completely in the noiseless case where (4) implies perfect estimation of the target with just one datapoint. To gain some intuition as to why having $\sigma^2 = 0$ hinders predictions of $\langle g^\star \rangle_{D_N}$ one can view it as a hard constraint ($f(x_n) = g(x_n)$), and hard constraints are typically less tractable than soft ones. Indeed this can also be seen as a motivation for considering the above Grand-canonical dataset ensemble. In a related view, finite $\sigma^2$ can be seen as a form of averaging which smooths and regulates analytical expressions making them more tractable. Another limitation of the EK result is that (to the best of our knowledge) there is no systematic way to extend it in orders of $1/N$ and get a more detailed picture of GPR generalization.

## 3.2 Field Theory Formulation

To facilitate the analysis of Eq. (1) we formulate the problem from a statistical-field-theory/path-integral viewpoint (Schulman, 1996). These are well-studied, powerful approaches for performing integrations over a space of functions (the jargon is "paths" when $x$ in one dimensional and "fields" when $x$ is higher dimensional). This section is somewhat technical and the reader who is not interested in derivations is invited to skip to section 3.3. To get some familiarity with this formalism, consider first averages over the (centered) GP itself with no dataset. Using the path-integral formalism we define the probability density functional of a GP with kernel $K$:

$$P_0[f] = \frac{\exp\left(-\frac{1}{2} \int dx dx' f(x) K^{-1}(x, x') f(x')\right)}{\int D\tilde{f} \exp\left(-\frac{1}{2} \int dx dx' \tilde{f}(x) K^{-1}(x, x') \tilde{f}(x')\right)} \tag{5}$$

where $\int Df$ denotes integration over the space of (well behaved) functions, for concreteness we limit $\int dx'$ to some compact domain such as the hyper-sphere, $K^{-1}(x, x')$ is the inverse covariance function ($\int dx' K(x, x') K^{-1}(x', x'') = \delta(x - x'')$). $P_0$ is analogous to a PDF with the sample space being the set of real functions. To define the path-integrals one first chooses an orthonormal basis of functions $\phi_i(x)$ (with respect to $\int dx$) arranged in order of likeliness $P_0[\phi_i] \geq P_0[\phi_j]$ for $i > j$ (note that this comparison doesn't require calculating the path integral in (5)). Second, one expands $f = \sum_i f_i \phi_i(x)$, and defines the path-integral as a series of simple integrals

$$\int Df \mathcal{F}[f] = \int df_1 \int df_2 \ldots \mathcal{F}\left[\sum_i f_i \phi_i\right] \tag{6}$$

where $\mathcal{F}$ is some functional of $f$ (like $P_0$ in (5)). Finally, one makes this last expression well-defined by taking a limit procedure where the number of integrals is gradually taken to infinity (Schulman, 1996).

Performing the above procedure we show in App. E that a stochastic process with functional probability density $P_0$ is equivalent to the original GP. Following a similar procedure, and denoting $\|f\|_K^2 = \int dx dx' f(x) K^{-1}(x, x') f(x')$ one can show an alternative representation of (1)

$$g^\star(x_\star) = Z^{-1} \int Df \cdot f(x_\star) \cdot \exp\left(-\frac{1}{2} \|f\|_K^2 - \frac{1}{2\sigma^2} \sum_{n=1}^N (f(x_n) - g_n)^2\right) \tag{7}$$

$$Z = \int Df \exp\left(-\frac{1}{2} \|f\|_K^2 - \frac{1}{2\sigma^2} \sum_{n=1}^N (f(x_n) - g_n)^2\right)$$

where $Z$ is known as the partition function (see Rasmussen & Williams (2005) for equivalence of (1) and (7)). It is useful to define the partition function with a "source term":

$$Z[\alpha(x)] = \int Df \exp\left(-\frac{1}{2} \|f\|_K^2 + \int dx \alpha(x) f(x) - \frac{1}{2\sigma^2} \sum_{n=1}^N (f(x_n) - g_n)^2\right) \tag{8}$$

Then the GPR prediction is simply

$$g^\star(x_\star) = \frac{\partial \log(Z[\alpha(x_\star)])}{\partial \alpha(x_\star)}\bigg|_{\alpha(x_\star)=0} \tag{9}$$

where $\frac{\partial \alpha(x)}{\partial \alpha(x_\star)} = \delta(x - x_\star)$. This form makes it clear that we can get quantities like dataset averaged predictions by finding the average of $\log Z$. By using the grand canonical ensemble, averaging over draws from the dataset can be carried using the "replica trick" (see for instance (Gardner & Derrida, 1988)), which aids in averaging over expressions like $\log(Z)$ and their derivatives via the equality $\log(Z) =_{M \to 0} \frac{Z^M - 1}{M}$. Specifically:

$$\langle g^\star(x_\star) \rangle_\eta = \left\langle \lim_{M \to 0} \frac{\partial}{\partial \alpha(x_\star)} \frac{Z^M - 1}{M}\bigg|_{\alpha(x_\star)=0} \right\rangle_\eta \tag{10}$$

$$\langle Z^M \rangle_\eta = \int \prod_{m=1}^{M} Df_m \exp\left(-\frac{1}{2}\sum_{m=1}^{M} \|f_m\|_K^2 + \eta \int d\mu_x e^{-\frac{1}{2\sigma^2}\sum_{m=1}^{M}(f_m(x)-g(x))^2}\right)$$

where, as standard in the replica formalism, the computation should be carried at positive integer $M$ and the analytical result extrapolated to zero at the end. Neatly, a Taylor expansion in $\eta$ of the above r.h.s. yields the $\langle ... \rangle_\eta$ averaging appearing on the l.h.s.

The main benefit of (10) over (1) is that it allows for a controlled expansion in $1/\eta$. At large $\eta$ (or similarly large $N$) we expect the fluctuations in $f_m(x)$ to be small and centered around $g(x)$. Indeed such a behavior is encouraged by the term multiplied by $\eta$ in the exponent. We can therefore systematically Taylor expand the inner exponent

$$e^{-\frac{\sum_{m=1}^{M}(f_m(x)-g(x))^2}{2\sigma^2}} = 1 - \frac{\sum_{m=1}^{M}(f_m(x)-g(x))^2}{2\sigma^2} + \frac{1}{2}\left[\frac{\sum_{m=1}^{M}(f_m(x)-g(x))^2}{2\sigma^2}\right]^2 + ... \tag{11}$$

and each term will yield a higher order of $\langle g^\star(x_\star) \rangle_\eta$ in $1/\eta$. The first order of the systematic expansion we defined can be dealt with in an exact manner, in which case we recovers the aforementioned EK result (4) with the slight difference that $N$ gets replaced with $\eta$ (see App. F for the computation). The second order term and further terms render the theory non-Gaussian and cannot be dealt with exactly but rather through the use of perturbation-theory/Feynman-diagrams.

### 3.3 FIELD THEORY FORMULATION PREDICTIONS

Performing the calculation described in the previous section we obtain the following closed expression for large $N$ or $\eta$ behavior of noisy GPR using any kernel, in particular kernels associated with deep fully-connected networks and convolutional networks.

$$\langle g^\star(x_\star) \rangle_\eta = g^\star_{EK,\eta}(x_\star) + g^\star_{SL,\eta}(x_\star) + O(1/\eta^3) \tag{12}$$

$$g^\star_{SL,\eta}(x_\star) = \frac{\eta}{\sigma^4}\sum_{i,j,k}\frac{\frac{\sigma^2}{\eta}}{\lambda_i + \frac{\sigma^2}{\eta}}\left(\frac{1}{\lambda_j} + \frac{\eta}{\sigma^2}\right)^{-1}\left(\frac{1}{\lambda_k} + \frac{\eta}{\sigma^2}\right)^{-1} g_i\phi_j(x_\star)\int d\mu_x \phi_i(x)\phi_j(x)\phi_k^2(x)$$

The first term is the familiar EK results but with $N$ replaced by $\eta$. The second term ($g^\star_{SL,\eta}$) is a sub-leading correction which captures the behavior at lower $N$'s.

As shown App. F similar expressions for $\langle g^{\star 2} \rangle_\eta$ are obtained using two replica indices. Interestingly we find that $\langle g^{\star 2} \rangle_\eta = \langle g^\star \rangle_\eta^2 + O(1/\eta^3)$. Hence the averaged MSE error is simply $(\langle g^\star(x_\star) \rangle_\eta - g(x_\star))^2$ integrated over $x_\star$. Since the variance of $g^\star$ came out to be $O(1/\eta^3)$ one finds that $g^\star - g$, which is $O(1/\eta)$, is asymptotically much larger than its standard deviation. This implies self averaging at large $\eta$, or equivalently that our dataset-averaged results capture the behavior of a single fixed dataset.

Equation (12) and the average MSE error are one of our key results. They provides us with closed expressions for the dataset-averaged MSE loss as a function of $\eta$ namely, the fixed-teacher learning curve. They hold without any limitations on the dataset or the kernel and yield a variant of the EK result along with its sub-leading correction. From an analytic perspective, once $\lambda_i$ and $\phi_i(x)$ are known, the above expressions provide clear insights to how well the GP learns each feature and what unwanted cross-talk is generated between features due to the second sub-leading term. Notably for the renormalized NTK introduced below, the number of non-zero $\lambda_i$'s is finite, and so the above infinite summations reduce to finite ones. This makes these expressions computationally superior to directly performing the matrix-inversion in (1) along with an $N-$dimensional integral involved in dataset-averaging. In addition having the sub-leading correction allows us to estimate the range of validity of our approximation by comparing the sub-leading and leading contributions, as we shall do for the uniform case below.

## 4 UNIFORM DATASETS

To make the result (12) interpretable, $\phi_i(x)$ and $\lambda_i$ are required. This can be done most readily for the case of datasets normalized to the hypersphere ($\|x_n\| = 1$) with a uniform probability measure and rotation-symmetric kernel functions. By the latter we mean $K_{x,x'} = K_{Ox,Ox'}$ for any $O$, where $O$ is an orthogonal matrix over the space of inputs. Although beyond the scope of the current work obvious extensions to consider are datasets which are uniform only in a sub-space of $x$ and/or small perturbations to uniformity.

Importantly, the NTK associated with any DNN with a fully connected first layer and weights initialized from a normal distribution, has the above symmetry under rotations. This follows from the recursion relations defining the NTK (Jacot et al., 2018) along with fact that the kernel of the first fully-connected layer is only a function of $x \cdot x'$. It follows that the NTK can be expanded as $K_{x,x'} = \sum_n b_n (x \cdot x')^n$. An additional corollary (Azevedo & Menegatto, 2015) is that its features are hyperspherical harmonics ($Y_{lm}(x)$) as these are the features of all dot product kernels. Hyperspherical harmonics are a complete (and orthonormal w.r.t a uniform measure) basis for functions on the hypersphere. For each $l$ these can be written as a sum of polynomials in the input coordinates of degree $l$. The extra index $m$ enumerates an orthogonal set of such polynomials (of size $\deg(l)$). [1] For a kernel of the above form the eigenvalues are independent of $m$ and given by (Azevedo & Menegatto, 2015)

$$\lambda_l = \frac{\Gamma\left(\frac{d}{2}\right)}{\sqrt{\pi} \cdot 2^l} \sum_{s=0}^{\infty} b_{2s+l} \frac{(2s+l)!}{(2s)!} \frac{\Gamma\left(s + \frac{1}{2}\right)}{\Gamma\left(s + l + \frac{d}{2}\right)} \tag{13}$$

For ReLU and erf activations, the $b_n$'s, can be obtained analytically up to any desirable order. Thus one can semi-analytically obtain the NTK eigenvalues up to any desired accuracy. For the particular case of depth 2 ReLU networks, we report in the App. H closed expression where the above summation can be carried out analytically. However as we shall argue soon, it is in fact desirable to trim the NTK in the sense of cutting-off its Taylor expansion at some order $m$, resulting in what we call the renormalized NTK. For such kernels, which would be our main focus next, the above result can be seen as a closed analytical expression for the eigenvalues.

Interestingly, for any fully-connected network and uniform datasets of dimension $d$ on the hypersphere, there is a universal bound given by $\lambda_l \leq K/\deg(l) \approx O(d^{-l})$, where $K$ is $K_{x,x}$ which is a constant in $x$. Indeed note that $K_{x,x}$ is finite and therefore its integral over the hypersphere is also finite and given by $\int d\mu_x K_{x,x} = K_{x,x} = \sum_{lm} \lambda_l = \sum_l \deg(l)\lambda_l$. The degeneracy ($\deg(l)$) is fixed from properties of hyper spherical harmonics, and equals $\deg(l) = \frac{2l+d-2}{l+d-2}\binom{l+d-2}{l}$ (Frye & Efthimiou, 2012) which goes as $O(d^l)$ for $l \ll d$. This combined with the positivity of the $\lambda_l$'s implies the above bound.

---

[1] Note that usually the hyperspherical harmonics are normalized w.r.t Lebesgue measure on the hypersphere, but in this context the normalization is w.r.t a probability measure on the hypersphere.

Expressing our target in this feature basis $g(x) = \sum_{l,m} g_{lm} Y_{lm}(x)$ (12) simplifies to

$$g^\star = g^\star_{EK,\eta}(x_\star) - \sum_{l,m} \left[ \frac{\eta^{-1} C_{K,\sigma^2/\eta}}{\lambda_l + \sigma^2/\eta} \frac{\lambda_l}{\lambda_l + \sigma^2/\eta} \right] g_{lm} Y_{lm}(x_\star) \qquad (14)$$

where $C_{K,\sigma^2/\eta} = \sum_{lm} (\lambda_l^{-1} + \eta/\sigma^2)^{-1}$ and notably cross-talk between features has been eliminated at this order since $\sum_m \phi_{lm}(x)^2$ is constant yielding $\sum_{\tilde{m}} \int d\mu_x \phi_{lm}(x) \phi_{l'm'}(x) \phi^2_{l\tilde{m}}(x) \propto \delta_{ll'} \delta_{mm'}$. By splitting the sum, $C_{K,\sigma^2/\eta}$, to cases in which $\lambda_l < \sigma^2/\eta$ and their complement one has the bound $C_{K,\sigma^2/\eta} < \#F\sigma^2/\eta + \sum_{lm|\lambda_l > \sigma^2/\eta} \lambda_l$, where $\#F$ is the number of non-zero kernel eigenvalues. Thus for kernels with a finite number of non-zero $\lambda_i$'s, as the renormalized NTK introduced below, $C_{K,\sigma^2/\eta}$ has a $\eta^{-1}$ asymptotic. This illustrates the fact the above terms are arranged by their orders in $\eta$.

Taking the leading order term one obtains the aforementioned EK result with $N$ replaced by $\eta$. Equating the two contributions provides an estimate of when perturbation theory breaks down. Focusing on $\lambda_l > \sigma^2/\eta$, the perturbation theory appears valid when $C_{K,\sigma^2/\eta} \ll \sigma^2$. In the limit $\sigma^2 \to 0$, and for trimmed kernels, this yield $\#F \ll \eta$. Notably it means that the original non-trimmed NTK cannot be analyzed perturbatively in the noiseless limit. In the next section we tackle this issue.

## 5 GENERALIZATION IN THE NOISELESS CASE AND THE RENORMALIZED NTK

The correspondence between DNNs trained in the NTK regime and GPR using NTK implies noiseless GPR ($\sigma^2 = 0$) for which the perturbative analysis carried in previous sections fails. Here we show that the fluctuations of $f_m(x)$ associated with low $\lambda$'s can be traded for noise on the fluctuations of $f_m(x)$ associated with high $\lambda$'s thereby making our perturbative analysis applicable. As shown in the previous section, the lower $\lambda$'s correspond to higher spherical harmonics and hence have higher Fourier components. We argue that these higher Fourier modes can be marginalized over in a controlled manner to generate both noise and corrections to the high $\lambda$'s. This is very much in spirit of the renormalization group technique common in physics wherein high Fourier modes are integrated over to generate changes (renormalization) of some parameters in the probability distribution of the low Fourier modes.

We begin by defining a set of renormalized NTKs. As argued, an NTK of any fully-connected DNN can be expanded as $K_{x,x'} = \sum_{q=0}^\infty b_q (x \cdot x')^q$. The renormalized NTK at scale $r$ is then simply $K^{(r)}_{x,x'} = \sum_{q=0}^r b_q (x \cdot x')^q$. Harmoniously with this notation we denote the prediction of GPR with the original kernel as $g^\star_\infty$. Our claim is that GPR with $K$ and a noise of $\sigma^2$ can be well approximated by GPR with $K^{(r)}$ and noise $\sigma^2 + \sigma_r^2$ (where $\sigma_r^2 = \sum_{q=r+1}^\infty b_q$), for sufficiently large $r$. Specifically that the discrepancy between GPR predictions scales as $O(\sqrt{N} d^{-(r+1)/2}/K_{x_1,x_1})$, where $d$ is the effective data-input dimension. As can be inferred from (13), the renormalized NTK at scale $r$ has zero eigenvalues for all spherical Harmonics with $l > r$. Thus, as advertised, these high Fourier modes have been removed from the problem. In a related manner trimming the Taylor expansion after $(x \cdot x')^r$ effectively reduces our angular resolution and coarse grains the fine angular features captured by these spherical Harmonics with $l > r$.

To justify this approximation we consider the difference matrix $K^{(r)}_{x_n,x_m} - K_{x_n,x_m}$, first for $n \neq m$ and $x_n$'s drawn for a uniform distribution of a hypersphere of dimension $d$. The terms $b_q(x_n \cdot x_m)^q$ scale roughly as $d^{q/2}$ (see App. H for a more accurate expression) due to the tendency of random vectors in high dimensions to be orthogonal. Consequently the above difference diminishes very quickly with $r$. Notably this also applies for $K^{(r)}_{x_*,x_m} - K_{x_*,x_n}$ provided $x_*$ is a test point and not a train point. In contrast the diagonal part of the different matrix is simply $\sigma_r^2 \delta_{n,m}$ and may diminish more slowly depending on the coefficients $b_{q>r}$. Upon neglecting $K^{(r)}_{x_*,x_m} - K_{x_*,x_n}$ and the off-diagonal elements of the difference matrix, one finds that (1) with these two GPRs yields identical predictions. As shown in App. H, these neglected off-diagonal elements yield a discrepancy which scales as $\sqrt{N} d^{-(r+1)/2}$ (since they sum incoherently). Consequently the MSE error between the two GPRs should scale as $N$. This scaling with $N$ should saturate when the accuracy is nearly perfect since then the predictions remain largely constant as $N$ is increased.

Focusing back on the question of how to tackle noiseless GPR, we thus find that as long as the $b_q$'s decays slowly enough with $q$, then at any finite $N$ we can choose a large enough $r$ such that two desirable properties are maintained: A. The discrepancy between the GPRs is small and B. $\sigma_r^2$ is large enough to ensure convergence to our perturbative analysis. The required slow decay of $b_q$ is harmonious with the intuition that DNNs should be initialized at the edge of chaos (Schoenholz et al., 2016) where the output of the network has a fine and multi-scale sensitivity to small changes in the input. As $K_{x,x'}$ is the correlation of two outputs with inputs $x$ and $x'$, having a power law decaying $b_q$ implies such fine and multi-scale sensitivity. Establishing relations between good initialization and effectiveness of our renormalized NTK is left for future work.

We have tested the accuracy of approximating noiseless NTK GPR with renormalized NTK GPR with the appropriate $\sigma_r^2$, both on artificial datasets (see next section) and on real world dataset such as CIFAR10 (see App B.). In both cases we found an excellent agreement between the two GPRs for $r$'s as small as 3 and 4.

## 6 GENERALIZATION IN THE NTK REGIME

Collecting the results of all the preceding sections, we can obtain a detailed and clear picture of generalization in fully connected DNNs trained in the NTK-regime on datasets with a uniform distribution normalized to some hypersphere in input space. We begin with a qualitative discussion and consider some renormalized NTK at scale $r$. From Sec. 4, we have that the features of this kernel are hyperspherical harmonics and that $\lambda_l$ scales as $d^{-l}$. We also recall that $l$ is the maximal degree of the polynomial appearing in the Harmonic and all Harmonics up to the degree $l$ span all polynomial on the hypersphere with degree up to $l$. Examining (14) we find that features with $\lambda_l \gg \sigma^2/\eta$ are learnable and via the above scaling we find that we learn polynomials of degree $O(\log(\eta/\sigma^2)/\log(d))$ or less. In particular a function like parity, which is a polynomial of degree $d$ is very hard to learn whereas a linear function is the easiest to learn. Thus despite having infinitely more parameters than data-points (due to infinite width) and despite being able to span almost any function (due to the richness of the kernel's features), deep neural networks avoid over fitting by having a strong bias towards low degree polynomials.

To make more quantitative statements we now focus on a specific setting. We consider input data in dimension $d = 50$ and a scalar target function $g(x) = \sum_{l=1,2;m} g_{lm} Y_{lm}(x)$ such that $\sum_{l=1,m} g_{lm}^2 = \sum_{l=2,m} g_{lm}^2 = 1/2$, but otherwise iid $g_{lm}$'s. We generate several toy datasets $D_N$ consisting of $N$ data points $(x_n)$ uniformly distributed on the hypersphere $S^{d-1}$ and their corresponding targets $(g(x_n))$. We consider training a fully-connected DNN consisting of 4 layer with ReLU activations and width $W$ which we initialize with variance $(\sigma_w^2 = \sigma_b^2 = 1/d)$ for the input layer and $(\sigma_w^2 = \sigma_b^2 = 1/W)$ for the hidden layers (see for instance Lee et al. (2019) App. C and App. E for how to compute the kernel). To be in the NTK correspondence regime we consider training such a network at vanishing learning-rate, MSE loss, and with $W \gg N$. One then has that the predictions of the DNN are given by GPR with $\sigma^2 = 0$ and the $K$ given by the NTK kernel (Jacot et al., 2018) [2].

For each such DNN we obtained the expected MSE loss $\|g_\infty^\star - g\|^2$ of GPR with the NTK kernel by numerical integration over $x_\star$ Repeating this process multiple times we obtained the dataset averaged loss $\langle \|g_\infty^\star - g\|^2 \rangle_{D_N}$ for $N = 1, 2, \ldots, N_{\max}$ with a relative standard error of less then 5% (this typically required averaging over 10 datasets). For direct comparison with our prediction of the learning curve, we computed the Poisson averaged learning curve $\langle \|g_\infty^\star - g\|^2 \rangle_\eta$ in accordance with (3), neglecting the terms $n > N_{\max}$. We restricted ourselves to $\eta_{\max} \le N_{\max} - 5\sqrt{N_{\max}}$ to make tail effects negligible. Notably the Poisson averaging makes the final statistical error negligible relative to the discrepancies coming from our large $\eta$ approximations (see A).

Since the target function involves $l = 0, 1, 2$ the minimal scale for the renormalized NTK is $r \ge 2$. To have some headroom we start from $r = 3$ which implies reasonable discrepancies of the MSE between the two GPRs of the order of $N/d^4 = 5.6e-4$, for the maximal $N$ we have ($N = 3500$). Our

---

[2]To be more precise, Jacot et al. (2018) predict correspondence with GPR up to a random initialization factor, so to get exact match with GPR one would also need to average over initialization seeds. Recent research (Lee et al., 2019) suggests this caveat can be avoided under some conditions.

analytical expressions following (13) combined with known results (Jacot et al., 2018; Cho & Saul, 2009) about the Taylor coefficients ($b_n$) yield $\lambda_0, ..., \lambda_3 = \{3.19, 7.27e-3, 5.98e-6, 1.62e-7\}$ and $\sigma_r^2 = 0.018$. Since $\lambda_0, \lambda_1 \gg \sigma^2/\eta \gg \lambda_2, \lambda_3$ for $50 < \eta < 3500$, $C_{K_r,\sigma^2/\eta}\sigma^{-2} < [\deg(0) + \deg(1)]\sigma^2/\eta + O(\deg(2)10^{-6})$, thus $C_{K_r,\sigma^2/\eta}\sigma^{-2} \approx 51/\eta$. Thus we expect perturbation theory to be valid for $\eta \gg 50$. At $\eta = 100$ the $l = 1$ features are learned well since $\sigma^2/\eta = 1.8e-4 \gg \lambda_1$ and the $l = 2$ features neglected, at $\eta = 1000$ they are learned but suppressed by a factor of about 3. Had the target contained $l = 3$ features, they would have been entirely neglected at these $\eta$ scale. Experimental learning curves along with our leading and sub-leading estimates are shown in Fig. 1. left panel showing an excellent agreement between theory and experiment.

While no actual DNNs were trained in the above experiments, the NTK correspondence means that this would be the exact behavior of a DNN trained in the NTK regime (Jacot et al., 2018; Lee et al., 2019; Arora et al., 2019). Furthermore since our aim was to estimate what the DNNs would predict rather than reach SOTA predictions, we focus on reasonable hyper-parameters but did not perform any hyper-parameter optimization.

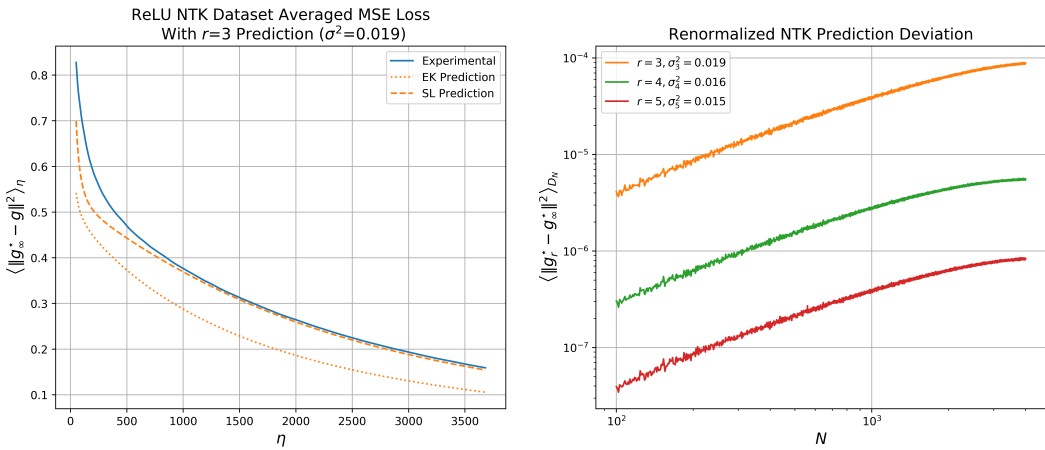

Figure 1: **Left panel:** The experimental learning curve (solid line) for a depth 4 ReLU network trained in the NTK regime on quadratic target function on a $d = 50$ hypersphere is shown along with our analytical predictions for the leading (dotted line) and leading plus sub-leading behavior (dashed line). **Right panel:** For the same dataset, we plot the dataset-averaged difference between predictions based on NTK ($g_\infty^\star$) and the renormalized NTK at scale $r$ ($g_r^\star$) showing an excellent agreement as $r$ increases.

Lastly we argue that the asymptotic behavior of learning-curve we predict is more accurate than the recent PAC based bounds (Allen-Zhu et al., 2018; Cao & Gu, 2019b;a). In App. C we show a log-log plot of the learning-curves contrasted with a $1/\sqrt{\eta}$ which is the most rapidly decaying bound appearing in those works. It can be seen that such an asymptotic cannot be made to fit the experimental learning-curve with any precision close to ours.

# 7 DISCUSSION AND OUTLOOK

In this work we laid out a formalism based on field theory tools for predicting learning-curves in the NTK regime. Well within the validly regime of our perturbative analysis we find excellent agreement to within $3\%$ relative mismatch between our best estimate and the experimental curves. Central to our analysis was the renormalization-group transformation on the NTK leading to effective observation noise on the target. Our analysis could be readily extend in several ways: Going beyond the uniform dataset case should be possible for multi-variate Gaussian input distribution with a set of similar finite variances and a set of nearly zero variances. Adding weak randomness to $K_{x,x'}$ to study the difference between empirical and averaged NTKs. It would also be interesting to extend our analysis to simple CNNs. The renormalized kernel can also be used for spectral analysis of the NTK and other kernels associated with DNNs.

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

## A    POISSON AVERAGING DEMONSTRATION

Here we demonstrate that Poisson averaging has no substantial effect on the learning curve. To this end we show the experimental learning curve from the main text pre- and post-averaging. It is evident that other than the unintended consequence of eliminating the experimental noise, the averaged learning curve is equivalent to the original for all practical intents.

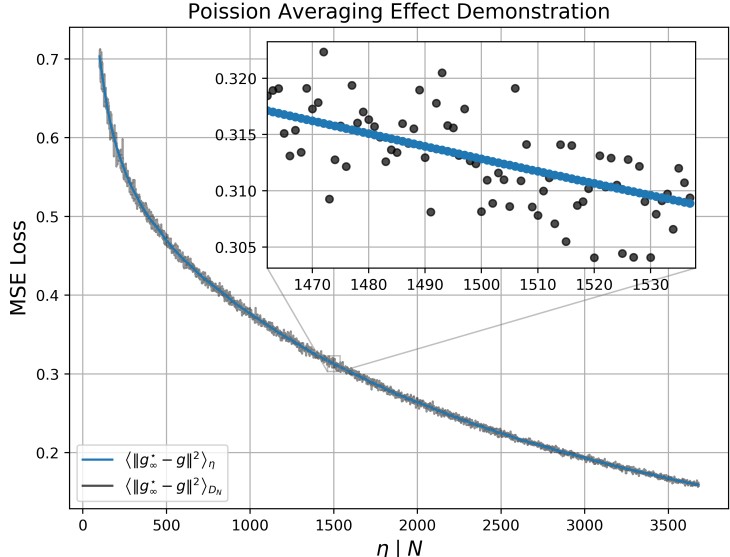

## B Comparison of NTK and Renormalized NTK Predictions on Non-Uniform Dataset

While our lack of knowledge of the NTK eigenvalues and eigenfunctions with respect to a non-uniform measure prevents us from predicting learning curves, we would like to show that the renormalized NTK is still a valid approximation in this setting. To this end we compare the prediction of the NTK and renormalized NTK on the one-hot encoding of the cifar-10 dataset.

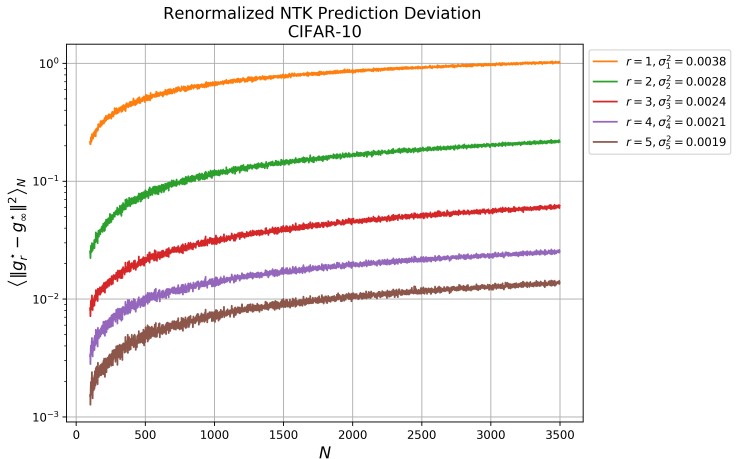

## C Comparison with Recent Bounds

As mentioned in the main text, various recent bounds, relevant to the NTK regime, have been derived recently. Notwithstanding importance and rigor of these works, their bounds have at best a $1/\sqrt{N}$ asymptotic scaling. Here we show that given a functional behavior of the experimental learning curves such a bound cannot be nearly as tight as our predictions.

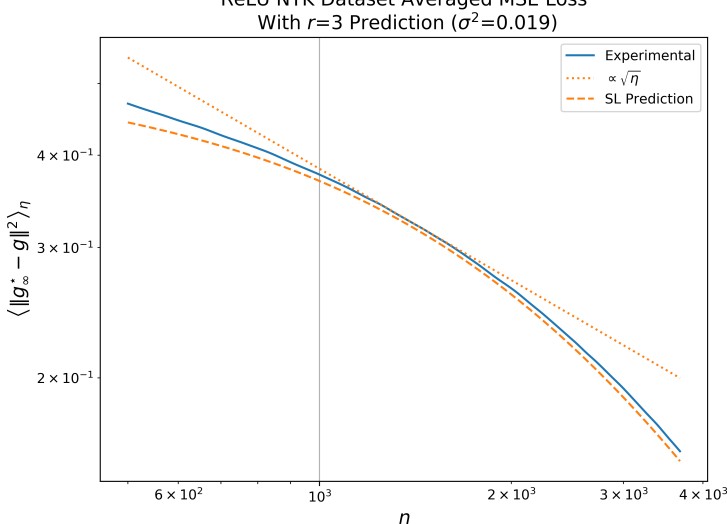

## D  NOTATIONS FOR THE FIELD THEORY DERIVATION.

For completeness, here we re-state the notations used in the main-text.

$\Omega$ - Space pace of inputs.

$x, x', x^*$ - Inputs (in $\Omega$).

$\mu_x$ - Measure on $\Omega$.

$K(x, x')$ - Kernel function (covariance) of a Gaussian process. Assumed to be symmetric and positive-semi-definite.

$\phi_i(x)$ - $i$'th eigenfunction of $K(x, x')$. By the spectral theorem, the set $\{\phi_i\}_{i=1}^{\infty}$ can be assumed to be orthonormal:

$$\int_{x\in\Omega} d\mu_x \phi_i(x) \phi_j(x) = \delta_{ij}$$

$\lambda_i$ - $i$'th eigenvalue of $K(x, x')$.

$$\int_{x'\in\Omega} d\mu_{x'} K(x, x') \phi_i(x') = \lambda_i \phi_i(x)$$

$||\cdot||_{\mathcal{H}_K}$ - RKHS norm. If $f(x) = \sum_i f_i \phi_i(x)$ then $||f||_{\mathcal{H}_K} = \sum_i \frac{f_i^2}{\lambda_i}$ (where $\phi_i$ is an orthonormal set). Note that this norm is independent of $\mu_x$.

$g(x)$ - The target function.

$\sigma^2$ - Noise variance. The noise is assumed to be Gaussian.

$N$ - Number of inputs in the data-set.

$D_N$ - Data-set of size $N$, $D_N = \{x_1, ..., x_N\}$.

$f^*_{D_N, \sigma^2}(x)$ - The prediction function.

## E  EXPLICIT PATH INTEGRAL COMPUTATIONS

Here we wish to prove the probability function defined in (5) of the main text yields the GP defined by a given kernel using explicit computation of the path integral. Denoting $\int d\mu_x d\mu_{x'} f(x) K^{-1}(x, x') f(x')$ as $||f||_K^2$ and noting that $||f||_K^2 = \sum_i \frac{f_i^2}{\lambda_i}$:

Notably, all other higher correlation functions split into products of the above correlation function due to standard properties of Gaussian integrals (Wick's/Isserlis' theorem).

$$\frac{\int Df \cdot f\left(x\right) \cdot f\left(y\right) \cdot \exp\left(-\frac{1}{2}\left\|f\right\|_K^2\right)}{\int Df \exp\left(-\frac{1}{2}\left\|f\right\|_K^2\right)} = \frac{\int \prod_i df_i \cdot \sum_i f_i\phi_i\left(x\right) \cdot \sum_j f_j\phi_j\left(y\right) \cdot \exp\left(-\frac{1}{2}\sum_l \frac{f_l^2}{\lambda_l}\right)}{\int \prod_i df_i \exp\left(-\frac{1}{2}\sum_l \frac{f_l^2}{\lambda_l}\right)} =$$

$$= \sum_i \underbrace{\frac{\int df \cdot f^2 \cdot \exp\left(-\frac{f^2}{2\lambda_i}\right)}{\int df \exp\left(-\frac{f^2}{2\lambda_i}\right)}}_{\lambda_i} \phi_i\left(x\right)\phi_i\left(y\right) + \sum_{i \neq j} \underbrace{\frac{\int df \cdot f \cdot \exp\left(-\frac{f^2}{2\lambda_i}\right)}{\int df \exp\left(-\frac{f^2}{2\lambda_i}\right)}}_{0} \cdot \underbrace{\frac{\int df \cdot f \cdot \exp\left(-\frac{f^2}{2\lambda_j}\right)}{\int df \exp\left(-\frac{f^2}{2\lambda_j}\right)}}_{0} =$$

$$= \sum_i \lambda_i\phi_i\left(x\right)\phi_i\left(y\right) = K\left(x,y\right)$$

## F    GAUSSIAN PROCESS PREDICTION AS A FIELD THEORY

Let us assume a Gaussian process (GP) with mean $0$ and co-variance function $K\left(x,x'\right)$. For a data-set $D_N$ of size $N$ and noisy targets ($\sigma^2 \neq 0$) $\{g\left(x_i\right)\}_{i=1}^N$, it is known that the posterior mean obtained by Bayesian inference is

$$f_{D_N,\sigma^2}^* = \arg\min_f \left[\frac{1}{2}\left\|f\right\|_{\mathcal{H}_K}^2 + \sum_{x_i \in D_N} \frac{\left(f\left(x_i\right) - g\left(x_i\right)\right)^2}{2\sigma^2}\right]$$

For a data-set $D_N$ of size $N$ and noisy targets $\{g\left(x_i\right)\}_{i=1}^N$ , we present the GP canonical partition function:

$$Z_{D_N,\sigma^2}\left[\alpha\left(x\right)\right] \stackrel{def}{=} \int Df \exp\left(-\frac{1}{2}\left\|f\right\|_{\mathcal{H}_K}^2 + \int \alpha\left(x\right)f\left(x\right)dx - \sum_{x_i \in D_N} \frac{\left(f\left(x_i\right) - g\left(x_i\right)\right)^2}{2\sigma^2}\right)$$

Where the $Df$ notation stands for path integral. Notice that the functional derivative of $\log\left(Z_{D_N,\sigma^2}\left[\alpha\left(x\right)\right]\right)$ w.r.t $\alpha\left(x^*\right)$ at $\alpha\left(x\right) = 0$ yields:

$$\left.\frac{\partial}{\partial\alpha\left(x^*\right)}\right|_{\alpha(x)=0} \log\left(Z_{D_N,\sigma^2}\left[\alpha\left(x\right)\right]\right) = \frac{1}{Z_{D_N,\sigma^2}\left[\alpha\left(x\right)=0\right]} \cdot \left.\frac{\partial}{\partial\alpha\left(x^*\right)}\right|_{\alpha(x)=0} \left(Z_{D_N,\sigma^2}\left[\alpha\left(x\right)\right]\right) =$$

$$= \frac{\int Df \cdot f\left(x^*\right)\exp\left(-\frac{1}{2}\left\|f\right\|_{\mathcal{H}_K}^2 - \sum_{x_i \in D_N} \frac{\left(f(x_i)-g(x_i)\right)^2}{2\sigma^2}\right)}{\int Df \exp\left(-\frac{1}{2}\left\|f\right\|_{\mathcal{H}_K}^2 - \sum_{x_i \in D_N} \frac{\left(f(x_i)-g(x_i)\right)^2}{2\sigma^2}\right)} = \arg\min_{f|_{x^*}}\left[\frac{1}{2}\left\|f\right\|_{\mathcal{H}_K}^2 + \sum_{i=1}^N \frac{\left(f\left(x_i\right) - g\left(x_i\right)\right)^2}{2\sigma^2}\right]$$

where the last equality is due to the fact that for Gaussian distributions, the expected value coincides with the most probable value. Therefore, the exact baysian inference mean:

$$f_{D_N,\sigma^2}^*\left(x^*\right) = \left.\frac{\partial}{\partial\alpha\left(x^*\right)}\right|_{\alpha(x)=0} \log\left(Z_{D_N,\sigma^2}\left[\alpha\left(x\right)\right]\right)$$

### F.1    CANONICAL ENSEMBLE FORMALISM

for evaluating the quality of a certain GP, we're interested in the average prediction for all the data-sets of size $N$, meaning:

$$f_{N,\sigma^2}^C (x^*) \overset{def}{=} \left\langle f_{D_N,\sigma^2}^* (x^*) \right\rangle_{D_N \sim \mu_x^N} = \int d\mu_{x_1} \int d\mu_{x_2} \ldots \int d\mu_{x_N} f_{D_N=\{x_1,\ldots,x_N\},\sigma^2}^* (x^*).$$

Using the replica trick we obtain:

$$f_{N,\sigma^2}^C (x^*) = \lim_{M \to 0} \frac{\partial}{\partial \alpha (x^*)} \bigg|_{\alpha(x)=0} \frac{\left\langle Z_{D_N,\sigma^2}^M [\alpha(x)] \right\rangle_{D_N \sim \mu_x^N} - 1}{M}$$

Now, let us calculate $\left\langle Z_{D_N,\sigma^2}^M [\alpha(x)] \right\rangle_{D_N \sim \mu_x^N}$ for an integer $M$:

$$Z_{D_N,\sigma^2}^M [\alpha(x)] = \underbrace{\int \int \ldots \int}_{M} \prod_{j=1}^M Df_j$$

$$\exp \left( -\frac{1}{2} \sum_{j=1}^M \|f_j\|_{\mathcal{H}_K}^2 + \sum_{j=1}^M \int \alpha(x) f_j(x) \, dx - \sum_{j=1}^M \sum_{x_i \in D_N} \frac{(f_j(x_i) - g(x_i))^2}{2\sigma^2} \right)$$

$$Z_{N,M,\sigma^2} = \left\langle Z_{D_N,\sigma^2}^M [\alpha(x)] \right\rangle_{D_N \sim \mu_x^N} = \int d\mu_{x_1} \cdots \int d\mu_{x_N} Z_{D_N,\sigma^2}^M [\alpha(x)] =$$

$$= \underbrace{\int \ldots \int}_{M} \prod_{j=1}^M Df_j \exp \left( -\frac{1}{2} \sum_{j=1}^M \|f_j\|_{\mathcal{H}_K}^2 + \sum_{j=1}^M \int \alpha(x) f_j(x) \, dx \right) \left\langle \exp \left( -\sum_{j=1}^M \frac{(f_j(x) - g(x))^2}{2\sigma^2} \right) \right\rangle_{x \sim \mu_x}^N$$

so

$$f_{N,\sigma^2}^C (x^*) = \lim_{M \to 0} \frac{\frac{\partial}{\partial \alpha(x^*)} \big|_{\alpha(x)=0} \left\langle Z_{D_N,\sigma^2}^M [\alpha(x)] \right\rangle_{D_N \sim \mu_x^N}}{M}$$

## F.2 GRAND CANONICAL ENSEMBLE FORMALISM

We now wish to allow fluctuations in the value of $N$, meaning averaging over $f_{N,\sigma^2}^C (x^*)$ for different values of $N$. The motivation is to simplify the calculations, while averaging around a relatively confined region of $N$s. Let us average the canonical prediction while weighting $N$ according to Poisson distribution with expected value $\eta$:

$$f_{\eta,\sigma^2}^{GC} (x^*) \overset{def}{=} \sum_{N=0}^{\infty} \frac{e^{-\eta} \eta^N}{N!} f_{N,\sigma^2}^C (x^*).$$

and defining:

$$\mathcal{Z}_{\eta,M,\sigma^2} [\alpha(x)] = \sum_{N=0}^{\infty} \frac{e^{-\eta} \eta^N}{N!} \left\langle Z_{D_N,\sigma^2}^M [\alpha(x)] \right\rangle_{D_N \sim \mu_x^N}$$

we get:

$$f_{\eta,\sigma^2}^{GC} (x^*) = \frac{\partial}{\partial \alpha(x^*)} \bigg|_{\alpha(x)=0} \lim_{M \to 0} \frac{\mathcal{Z}_{\eta,M,\sigma^2} [\alpha(x)]}{M}$$

That is, the functional derivative w.r.t $\alpha\left(x^*\right)$ at $\alpha\left(x\right)=0$ yields the average prediction, averaged over different data-set sizes (the canonical averaging) and different data-sets for each size (the grand canonical averaging).

For a given $\eta$, the standard deviation of $N$ is $\eta$, so the relative error is $\frac{1}{\sqrt{\eta}}$, decreases with $\eta$.

Substituting $\left\langle Z_{D_N,\sigma^2}^M\left[\alpha\left(x\right)\right]\right\rangle_{D_N\sim\mu_x^N}$ in the expression for $\mathcal{Z}_{\eta,M,\sigma^2}\left[\alpha\left(x\right)\right]$ we obtain:

$$\left\langle Z_{D_N,\sigma^2}^M\left[\alpha\left(x\right)\right]\right\rangle_{D_N\sim\mu_x^N}=$$

$$=\sum_{N=0}^{\infty}\frac{e^{-\eta}\eta^N}{N!}\underbrace{\int...\int}_{M}\prod_{j=1}^M Df_j\exp\left(-\frac{1}{2}\sum_{j=1}^M\|f_j\|_{\mathcal{H}_K}^2+\sum_{j=1}^M\int\alpha\left(x\right)f_j\left(x\right)dx\right)$$

$$\left\langle\exp\left(-\sum_{j=1}^M\frac{\left(f_j\left(x\right)-g\left(x\right)\right)^2}{2\sigma^2}\right)\right\rangle_{x\sim\mu_x}^N=$$

$$=\underbrace{\int...\int}_{M}\prod_{j=1}^M Df_j\exp\left(-\frac{1}{2}\sum_{j=1}^M\|f_j\|_{\mathcal{H}_K}^2+\sum_{j=1}^M\int\alpha\left(x\right)f_j\left(x\right)dx\right)$$

$$\sum_{N=0}^{\infty}\frac{e^{-\eta}\eta^N}{N!}\left\langle\exp\left(-\sum_{j=1}^M\frac{\left(f_j\left(x\right)-g\left(x\right)\right)^2}{2\sigma^2}\right)\right\rangle_{x\sim\mu_x}^N=$$

$$=e^{-\eta}\underbrace{\int...\int}_{M}Df_1...Df_M$$

$$\exp\left(-\frac{1}{2}\sum_{j=1}^M\|f_j\|_{\mathcal{H}_K}^2+\sum_{j=1}^M\int\alpha\left(x\right)f_j\left(x\right)dx+\eta\left\langle\exp\left(-\sum_{j=1}^M\frac{\left(f_j\left(x\right)-g\left(x\right)\right)^2}{2\sigma^2}\right)\right\rangle_{x\sim\mu_x}\right)$$

## F.3 DERIVING THE EQUIVALENCE KERNEL USING THE GRAND CANONICAL FORMALISM

We wish to get rid of the exponent inside the exponent. Expending it using (first order) Taylor series:

$$\mathcal{Z}_{\eta,M,\sigma^2}\left[\alpha\left(x\right)\right] =$$

$$= e^{-\eta} \underbrace{\int ... \int}_{M} \prod_{j=1}^{M} Df_j$$

$$\exp\left(-\frac{1}{2}\sum_{j=1}^{M}\|f_j\|_{\mathcal{H}_K}^2 + \sum_{j=1}^{M}\int \alpha\left(x\right)f_j\left(x\right)dx + \eta\left\langle \exp\left(-\sum_{j=1}^{M}\frac{\left(f_j\left(x\right)-g\left(x\right)\right)^2}{2\sigma^2}\right)\right\rangle_{x\sim\mu_x}\right) \approx$$

$$\approx e^{-\eta} \underbrace{\int ... \int}_{M} \prod_{j=1}^{M} Df_j$$

$$\exp\left(-\frac{1}{2}\sum_{j=1}^{M}\|f_j\|_{\mathcal{H}_K}^2 + \sum_{j=1}^{M}\int \alpha\left(x\right)f_j\left(x\right)dx + \eta\left\langle 1 - \sum_{j=1}^{M}\frac{\left(f_j\left(x\right)-g\left(x\right)\right)^2}{2\sigma^2}\right\rangle_{x\sim\mu_x}\right) =$$

$$= \left[\int Df \exp\left(-\frac{1}{2}\|f\|_{\mathcal{H}_K}^2 + \int \alpha\left(x\right)f\left(x\right)dx - \eta\left\langle\frac{\left(f\left(x\right)-g\left(x\right)\right)^2}{2\sigma^2}\right\rangle_{x\sim\mu_x}\right)\right]^M \overset{def}{=} \left(\mathcal{Z}_{\eta,\sigma^2}^{EK}\left[\alpha\left(x\right)\right]\right)^M$$

$$\mathcal{Z}_{\eta,\sigma^2}^{EK}\left[\alpha\left(x\right)\right] = \int Df \exp\left(-\frac{1}{2}\|f\|_{\mathcal{H}_K}^2 + \int \alpha\left(x\right)f\left(x\right)d\mu_x - \frac{\eta}{2\sigma^2}\int d\mu_x\left(f\left(x\right)-g\left(x\right)\right)^2\right)$$

and without any additional approximation:

$$f_{\eta,\sigma^2}^{GC}\left(x^*\right) = \left.\frac{\partial}{\partial\alpha\left(x^*\right)}\right|_{\alpha(x)=0}\lim_{M\to 0}\frac{\mathcal{Z}_{\eta,M,\sigma^2}\left[\alpha\left(x\right)\right]-1}{M} \approx \left.\frac{\partial}{\partial\alpha\left(x^*\right)}\right|_{\alpha(x)=0}\lim_{M\to 0}\frac{\left(\mathcal{Z}_{\eta,\sigma^2}^{EK}\left[\alpha\left(x\right)\right]\right)^M-1}{M} =$$

$$= \left.\frac{\partial}{\partial\alpha\left(x^*\right)}\right|_{\alpha(x)=0}\log\left(\mathcal{Z}_{\eta,\sigma^2}^{EK}\left[\alpha\left(x\right)\right]\right) = \frac{1}{\mathcal{Z}_{\eta,\sigma^2}^{EK}\left[\alpha\left(x\right)=0\right]}\cdot\left.\frac{\partial}{\partial\alpha\left(x^*\right)}\right|_{\alpha(x)=0}\left(\mathcal{Z}_{\eta,\sigma^2}^{EK}\left[\alpha\left(x\right)\right]\right) =$$

$$= \frac{1}{\mathcal{Z}_{\eta,\sigma^2}^{EK}\left[\alpha\left(x\right)=0\right]}\cdot\int Df\cdot f\left(x^*\right)$$

$$\exp\left(-\frac{1}{2}\|f\|_{\mathcal{H}_K}^2 + \int \alpha\left(x\right)f\left(x\right)dx - \frac{\eta}{2\sigma^2}\int d\mu_x\left(f\left(x\right)-g\left(x\right)\right)^2\right)_{\alpha(x)=0} =$$

$$= \frac{\int Df\cdot f\left(x^*\right)\exp\left(-\frac{1}{2}\|f\|_{\mathcal{H}_K}^2 - \frac{\eta}{2\sigma^2}\int d\mu_x\left(f\left(x\right)-g\left(x\right)\right)^2\right)}{\int Df\exp\left(-\frac{1}{2}\|f\|_{\mathcal{H}_K}^2 - \frac{\eta}{2\sigma^2}\int d\mu_x\left(f\left(x\right)-g\left(x\right)\right)^2\right)} =$$

$$= \arg\min_{f|_{x^*}}\left[\frac{1}{2}\|f\|_{\mathcal{H}_K}^2 + \frac{\eta}{2\sigma^2}\int d\mu_x\left(f\left(x\right)-g\left(x\right)\right)^2\right] \overset{def}{=} f_{\eta,\sigma^2}^{EK}\left(x^*\right)$$

and that is exactly the result for the equivalence kernel, where $\eta$ is the data-set size (we regarded it as the mean of the data-set size).

Let us derive it explicitly. For $f\left(x\right) = \sum_i f_i\phi_i\left(x\right)$ and $g\left(x\right) = \sum_i g_i\phi_i\left(x\right)$:

$$f_{\eta,\sigma^2}^{EK}(x^*) = \frac{\int Df \cdot f(x^*) \exp\left(-\frac{1}{2}\|f\|_{\mathcal{H}_K}^2 - \frac{\eta}{2\sigma^2}\int d\mu_x \left(f(x) - g(x)\right)^2\right)}{\int Df \exp\left(-\frac{1}{2}\|f\|_{\mathcal{H}_K}^2 - \frac{\eta}{2\sigma^2}\int d\mu_x \left(f(x) - g(x)\right)^2\right)} =$$

$$= \frac{\int \prod_i df_i \cdot \sum_i f_i \phi_i(x^*) \cdot \exp\left(-\frac{1}{2}\sum_i \left(\frac{f_i^2}{\lambda_i} + \frac{\eta}{\sigma^2}(f_i - g_i)^2\right)\right)}{\int \prod_i df_i \exp\left(-\frac{1}{2}\sum_i \left(\frac{f_i^2}{\lambda_i} + \frac{\eta}{\sigma^2}(f_i - g_i)^2\right)\right)} =$$

$$= \sum_i \phi_i(x^*) \frac{\int df_i \cdot f_i \cdot \exp\left(-\frac{f_i^2}{2\lambda_i} - \frac{\eta}{2\sigma^2}(f_i - g_i)^2\right)}{\int df_i \exp\left(-\frac{f_i^2}{2\lambda_i} - \frac{\eta}{2\sigma^2}(f_i - g_i)^2\right)} = \sum_i \frac{\lambda_i}{\lambda_i + \frac{\sigma^2}{\eta}} g_i \phi_i(x^*)$$

## F.4 EQUIVALENCE KERNEL AS FREE FIELD THEORY

Regarding the Equivalence Kernel as the free (quadratic) theory, we can denote $f_{\eta,\sigma^2}^{EK}(x^*) = \langle f(x^*)\rangle_{f\sim EK} = \langle f(x^*)\rangle_0 = \sum_i \frac{\lambda_i}{\lambda_i + \frac{\sigma^2}{\eta}} g_i \phi_i(x^*).$ '

Let us calculate the correlations in the free theory:

$$\langle f(x) f(y)\rangle_0 = \frac{\int Df \cdot f(x) f(y) \exp\left(-\frac{1}{2}\|f\|_{\mathcal{H}_K}^2 - \frac{\eta}{2\sigma^2}\int d\mu_x \left(f(x) - g(x)\right)^2\right)}{\int Df \exp\left(-\frac{1}{2}\|f\|_{\mathcal{H}_K}^2 - \frac{\eta}{2\sigma^2}\int d\mu_x \left(f(x) - g(x)\right)^2\right)} =$$

$$= \frac{\int \prod_i df_i \cdot \sum_{i,j} f_i f_j \phi_i(x) \phi_j(y) \cdot \exp\left(-\frac{1}{2}\sum_i \left(\frac{f_i^2}{\lambda_i} + \frac{\eta}{\sigma^2}(f_i - g_i)^2\right)\right)}{\int \prod_i df_i \exp\left(-\frac{1}{2}\sum_i \left(\frac{f_i^2}{\lambda_i} + \frac{\eta}{\sigma^2}(f_i - g_i)^2\right)\right)} =$$

$$= \sum_i \underbrace{\frac{\int df_i \cdot f_i^2 \cdot \exp\left(-\frac{f_i^2}{2\lambda_i} - \frac{\eta}{2\sigma^2}(f_i - g_i)^2\right)}{\int df_i \exp\left(-\frac{f_i^2}{2\lambda_i} - \frac{\eta}{2\sigma^2}(f_i - g_i)^2\right)}}_{\frac{\lambda_i^2 g_i^2}{\left(\lambda_i + \frac{\sigma^2}{\eta}\right)^2} + \left(\frac{1}{\lambda_i} + \frac{\eta}{\sigma^2}\right)^{-1}} \phi_i(x) \phi_i(y) + \sum_{i\neq j} \frac{\lambda_i g_i \lambda_j g_j}{\left(\lambda_i + \frac{\sigma^2}{\eta}\right)\left(\lambda_j + \frac{\sigma^2}{\eta}\right)} \phi_i(x) \phi_j(y) =$$

$$= \sum_i \left(\frac{1}{\lambda_i} + \frac{\eta}{\sigma^2}\right)^{-1} \phi_i(x) \phi_i(y) + \underbrace{\sum_{i,j} \frac{\lambda_i g_i \lambda_j g_j}{\left(\lambda_i + \frac{\sigma^2}{\eta}\right)\left(\lambda_j + \frac{\sigma^2}{\eta}\right)} \phi_i(x) \phi_j(y)}_{\langle f(x)\rangle_0 \langle f(y)\rangle_0}$$

Therefore:

$$\mathrm{Cov}\left[f(x), f(y)\right] = \sum_i \left(\frac{1}{\lambda_i} + \frac{\eta}{\sigma^2}\right)^{-1} \phi_i(x) \phi_i(y)$$

and we see that the correlations are $O\left(\frac{1}{\eta}\right)$.

For rotationally invariant kernel, we get that

$$\mathrm{Var}\left[f(x)\right] = \sum_{l=0}^{\infty} \sum_{m=0}^{\deg(l)} \left(\frac{1}{\lambda_l} + \frac{\eta}{\sigma^2}\right)^{-1} Y_{l,m}^2(x) \stackrel{def}{=} C_{K,\eta,\sigma^2}$$

is independent of x since

$$\sum_{m=0}^{\deg(l)} Y_{l,m}^2(x) = \deg(l)$$

so

$$C_{K,\eta,\sigma^2} = \sum_{l=0}^{\infty} \sum_{m=0}^{\deg(l)} \frac{1}{\lambda_l^{-1} + \eta/\sigma^2}$$

## F.5 Pertubative Correction for the Equivalence Kernel

### F.5.1 Averaging $f$

Going to the next order in the expansion:

$$\mathcal{Z}_{\eta,M,\sigma^2}[\alpha(x)] =$$

$$= e^{-\eta} \underbrace{\int \cdots \int}_{M} \prod_{j=1}^{M} Df_j$$

$$\exp\left(-\frac{1}{2}\sum_{j=1}^{M}\|f_j\|_{\mathcal{H}_K}^2 + \sum_{j=1}^{M}\int \alpha(x)f_j(x)\,dx + \eta\left\langle \exp\left(-\sum_{j=1}^{M}\frac{(f_j(x)-g(x))^2}{2\sigma^2}\right)\right\rangle_{x\sim\mu_x}\right) \approx$$

$$\approx e^{-\eta} \underbrace{\int \cdots \int}_{M} \prod_{j=1}^{M} Df_j$$

$$\exp\left(-\frac{1}{2}\sum_{j=1}^{M}\|f_j\|_{\mathcal{H}_K}^2 + \sum_{j=1}^{M}\int \alpha(x)f_j(x)\,dx\right)$$

$$\exp\left(\eta\left\langle 1 - \sum_{j=1}^{M}\frac{(f_j(x)-g(x))^2}{2\sigma^2} + \frac{1}{2}\left(\sum_{j=1}^{M}\frac{(f_j(x)-g(x))^2}{2\sigma^2}\right)^2\right\rangle_{x\sim\mu_x}\right) =$$

$$= \underbrace{\int \cdots \int}_{M} \prod_{j=1}^{M} Df_j$$

$$\exp\left(\sum_{j=1}^{M}\left(-\frac{1}{2}\|f_j\|_{\mathcal{H}_K}^2 + \int \alpha(x)f_j(x)\,dx - \frac{\eta}{2\sigma^2}\int d\mu_x\,(f_j(x)-g(x))^2\right)\right)$$

$$\exp\left(\frac{\eta}{8\sigma^4}\sum_{j=1}^{M}\sum_{l=1}^{M}\int d\mu_x\,(f_j(x)-g(x))^2 \cdot (f_l(x)-g(x))^2\right)$$

Note that:

$$f_{\eta,\sigma^2}^{GC}(x^*) = \left.\frac{\partial}{\partial\alpha(x^*)}\right|_{\alpha(x)=0} \lim_{M\to 0} \frac{\mathcal{Z}_{\eta,M,\sigma^2}[\alpha(x)] - 1}{M} =$$

$$= \lim_{M\to 0}\left[\left(\mathcal{Z}_{\eta,\sigma^2}^{EK}[\alpha(x)=0]\right)^M \cdot \frac{\left.\frac{\partial}{\partial\alpha(x^*)}\right|_{\alpha(x)=0}\mathcal{Z}_{\eta,M,\sigma^2}[\alpha(x)]}{\left(\mathcal{Z}_{\eta,\sigma^2}^{EK}[\alpha(x)=0]\right)^M}\cdot\frac{1}{M}\right] =$$

$$= \underbrace{\lim_{M\to 0}\left(\mathcal{Z}_{\eta,\sigma^2}^{EK}[\alpha(x)=0]\right)^M}_{1} \cdot \lim_{M\to 0}\left[\frac{\left.\frac{\partial}{\partial\alpha(x^*)}\right|_{\alpha(x)=0}\mathcal{Z}_{\eta,M,\sigma^2}[\alpha(x)]}{\left(\mathcal{Z}_{\eta,\sigma^2}^{EK}[\alpha(x)=0]\right)^M}\cdot\frac{1}{M}\right]$$

Calculating the first order pertubative corrections:

$$f_{\eta,\sigma^2}^{GC}(x^*) = \lim_{M\to 0} \frac{\left.\frac{\partial}{\partial\alpha(x^*)}\right|_{\alpha(x)=0} \mathcal{Z}_{\eta,M,\sigma^2}[\alpha(x)]}{M\cdot\left(\mathcal{Z}_{\eta,\sigma^2}^{EK}[\alpha(x)=0]\right)^M} =$$

$$= \lim_{M\to 0} \frac{1}{M\mathcal{Z}_{\eta,\sigma^2}^{EK}[0]^M} \underbrace{\int \cdots \int}_{M} \prod_{j=1}^{M} Df_j$$

$$\exp\left(-\frac{1}{2}\sum_{j=1}^{M}\|f_j\|_{\mathcal{H}_K}^2 - \frac{\eta}{2\sigma^2}\sum_{j=1}^{M}\int d\mu_x\,(f_j(x)-g(x))^2\right)$$

$$\exp\left(\frac{\eta}{8\sigma^4}\sum_{j=1}^{M}\sum_{l=1}^{M}\int d\mu_x\,(f_j(x)-g(x))^2\cdot(f_l(x)-g(x))^2\right)\cdot\sum_{i=1}^{M}f_i(x^*) =$$

$$= \lim_{M\to 0} \frac{1}{M\mathcal{Z}_{\eta,\sigma^2}^{EK}[0]^M} \underbrace{\int \cdots \int}_{M} \prod_{j=1}^{M} Df_j \exp\left(-\frac{1}{2}\sum_{j=1}^{M}\|f_j\|_{\mathcal{H}_K}^2 - \frac{\eta}{2\sigma^2}\sum_{j=1}^{M}\int d\mu_x\,(f_j(x)-g(x))^2\right)$$

$$\left(1 + \frac{\eta}{8\sigma^4}\sum_{j=1}^{M}\sum_{l=1}^{M}\int d\mu_x\,(f_j(x)-g(x))^2\cdot(f_l(x)-g(x))^2\right)\cdot\sum_{i=1}^{M}f_i(x^*) + O\left(\frac{1}{\eta^3}\right) =$$

$$= \lim_{M\to 0} \frac{1}{M}\left\langle\left(\left(1 + \frac{\eta}{8\sigma^4}\sum_{j=1}^{M}\sum_{l=1}^{M}\int d\mu_x\,(f_j(x)-g(x))^2\cdot(f_l(x)-g(x))^2\right)\cdot\sum_{i=1}^{M}f_i(x^*)\right\rangle_{f_1,\ldots,f_M\sim EK}$$

$$+ O\left(\frac{1}{\eta^3}\right) =$$

$$= \langle f(x^*)\rangle_0 + \lim_{M\to 0}\frac{1}{M}\left\langle\left(\left(\frac{\eta}{8\sigma^4}\sum_{j=1}^{M}\sum_{l=1}^{M}\int d\mu_x\,(f_j(x)-g(x))^2\cdot(f_l(x)-g(x))^2\right)\cdot\sum_{i=1}^{M}f_i(x^*)\right\rangle_{f_1,\ldots,f_M\sim EK}$$

$$+ O\left(\frac{1}{\eta^3}\right) =$$

$$= \langle f(x^*)\rangle_0$$

$$+ \lim_{M\to 0}\frac{1}{M}\frac{\eta}{8\sigma^4}\int d\mu_x \left\langle\sum_{j=1}^{M}\sum_{l=1}^{M}\sum_{i=1}^{M}(f_j(x)-g(x))^2\cdot(f_l(x)-g(x))^2 f_i(x^*)\right\rangle_{f_1,\ldots,f_M\sim EK} + O\left(\frac{1}{\eta^3}\right).$$

Calculating the correction:

$$\left\langle\sum_{j=1}^{M}\sum_{l=1}^{M}\sum_{i=1}^{M}(f_j(x)-g(x))^2\cdot(f_l(x)-g(x))^2 f_i(x^*)\right\rangle_{f_1,\ldots,f_M\sim EK} =$$

$$= M\left\langle(f(x)-g(x))^4 f(x^*)\right\rangle_0$$

$$+ M(M-1)\left[2\left\langle(f(x)-g(x))^2\right\rangle_0\left\langle(f(x)-g(x))^2 f(x^*)\right\rangle_0 + \left\langle(f(x)-g(x))^4\right\rangle_0\langle f(x^*)\rangle_0\right]$$

$$+ M(M-1)(M-2)\left\langle(f(x)-g(x))^2\right\rangle_0^2\langle f(x^*)\rangle_0$$

Note that we eventually divide by $M$ and take the limit $M\to 0$, so we only care about $O(M)$ terms:

$$f^{GC}_{\eta,\sigma^2}(x^*) = f^{EK}_{\eta,\sigma^2}(x^*)$$

$$+\frac{\eta}{8\sigma^4}\int d\mu_x \left[ \left\langle (f(x)-g(x))^4 f(x^*) \right\rangle_0 - 2\left\langle (f(x)-g(x))^2 \right\rangle_0 \left\langle (f(x)-g(x))^2 f(x^*) \right\rangle_0 \right.$$

$$\left. - \left\langle (f(x)-g(x))^4 \right\rangle_0 \langle f(x^*) \rangle_0 + 2\left\langle (f(x)-g(x))^2 \right\rangle_0^2 \langle f(x^*) \rangle_0 \right] + O\left(\frac{1}{\eta^3}\right)$$

These correlations can be calculated using Feynman diagrams, since the free theory (EK) is quadratic (Gaussian):

$$\left\langle (f(x)-g(x))^4 f(x^*) \right\rangle_0 =$$

$$= 3f^{EK}_{\eta,\sigma^2}(x^*)\,\mathrm{Var}\,[f(x)]^2 + 6f^{EK}_{\eta,\sigma^2}(x^*)\left(f^{EK}_{\eta,\sigma^2}(x)-g(x)\right)^2 \mathrm{Var}\,[f(x)]$$

$$+f^{EK}_{\eta,\sigma^2}(x^*)\left(f^{EK}_{\eta,\sigma^2}(x)-g(x)\right)^4 + 4\left(f^{EK}_{\eta,\sigma^2}(x)-g(x)\right)^3 \mathrm{Cov}\,[f(x),f(x^*)]$$

$$+12\left(f^{EK}_{\eta,\sigma^2}(x)-g(x)\right)\mathrm{Var}\,[f(x)]\,\mathrm{Cov}\,[f(x),f(x^*)]$$

$$\left\langle (f(x)-g(x))^4 \right\rangle_0 \langle f(x^*) \rangle_0 =$$

$$= 3f^{EK}_{\eta,\sigma^2}(x^*)\,\mathrm{Var}\,[f(x)]^2$$

$$+6f^{EK}_{\eta,\sigma^2}(x^*)\left(f^{EK}_{\eta,\sigma^2}(x)-g(x)\right)^2 \mathrm{Var}\,[f(x)] + f^{EK}_{\eta,\sigma^2}(x^*)\left(f^{EK}_{\eta,\sigma^2}(x)-g(x)\right)^4$$

$$\left\langle (f(x)-g(x))^2 \right\rangle_0 \left\langle (f(x)-g(x))^2 f(x^*) \right\rangle_0 =$$

$$= 2\mathrm{Var}\,[f(x)]\,\mathrm{Cov}\,[f(x),f(x^*)]\left(f^{EK}_{\eta,\sigma^2}(x)-g(x)\right) + f^{EK}_{\eta,\sigma^2}(x^*)\mathrm{Var}\,[f(x)]^2$$

$$+2\mathrm{Cov}\,[f(x),f(x^*)]\left(f^{EK}_{\eta,\sigma^2}(x)-g(x)\right)^3$$

$$+2f^{EK}_{\eta,\sigma^2}(x^*)\mathrm{Var}\,[f(x)]\left(f^{EK}_{\eta,\sigma^2}(x)-g(x)\right)^2 + f^{EK}_{\eta,\sigma^2}(x^*)\left(f^{EK}_{\eta,\sigma^2}(x)-g(x)\right)^4$$

$$\left\langle (f(x)-g(x))^2 \right\rangle_0^2 \langle f(x^*) \rangle_0 =$$

$$= f^{EK}_{\eta,\sigma^2}(x^*)\,\mathrm{Var}\,[f(x)]^2$$

$$+2f^{EK}_{\eta,\sigma^2}(x^*)\mathrm{Var}\,[f(x)]\cdot\left(f^{EK}_{\eta,\sigma^2}(x)-g(x)\right)^2 + f^{EK}_{\eta,\sigma^2}(x^*)\left(f^{EK}_{\eta,\sigma^2}(x)-g(x)\right)^4$$

Summing everything up:

$$\left\langle (f(x)-g(x))^4 f(x^*) \right\rangle_0 - \left\langle (f(x)-g(x))^4 \right\rangle_0 \langle f(x^*) \rangle_0$$

$$-2\left\langle (f(x)-g(x))^2 \right\rangle_0 \left\langle (f(x)-g(x))^2 f(x^*) \right\rangle_0 + 2\left\langle (f(x)-g(x))^2 \right\rangle_0^2 \langle f(x^*) \rangle_0 =$$

$$= 8\left(f^{EK}_{\eta,\sigma^2}(x)-g(x)\right)\mathrm{Var}\,[f(x)]\,\mathrm{Cov}\,[f(x),f(x^*)]$$

and all the bubble diagrams cancel as expected.

So we get:

$$f_{\eta,\sigma^2}^{GC}(x^*) = f_{\eta,\sigma^2}^{EK}(x^*) + \frac{\eta}{\sigma^4} \int d\mu_x \left( f_{\eta,\sigma^2}^{EK}(x) - g(x) \right) \mathrm{Var}\left[ f(x) \right] \mathrm{Cov}\left[ f(x), f(x^*) \right] + O\left( \frac{1}{\eta^3} \right)$$

Substituting the expressions for the variance and covariance:

$$f_{\eta,\sigma^2}^{GC}(x^*) =$$

$$= f_{\eta,\sigma^2}^{EK}(x^*) - \frac{\eta}{\sigma^4} \sum_{i,j,k} \frac{\frac{\sigma^2}{\eta}}{\lambda_i + \frac{\sigma^2}{\eta}} \left( \frac{1}{\lambda_j} + \frac{\eta}{\sigma^2} \right)^{-1} \left( \frac{1}{\lambda_k} + \frac{\eta}{\sigma^2} \right)^{-1} g_i \phi_j(x^*) \int d\mu_x \phi_i(x) \phi_j(x) \phi_k^2(x) + O\left( \frac{1}{\eta^3} \right)$$

### F.5.2   AVERAGING $f^2$

This time we must use two different replica indices:

$$\left\langle \left[ f_{D_N,\sigma^2}^*(x^*) \right]^2 \right\rangle_{D_N \sim \mu_x^N} = \left\langle \left[ \left. \frac{\partial \log \left( Z_{D_N,\sigma^2}[\alpha(x)] \right)}{\partial \alpha(x^*)} \right|_{\alpha(x)=0} \right]^2 \right\rangle_{D_N \sim \mu_x^N} =$$

$$= \left\langle \left( \lim_{M \to 0} \frac{1}{M} \cdot \left. \frac{\partial Z_{D_N,\sigma^2}^M[\alpha(x)]}{\partial \alpha(x^*)} \right|_{\alpha(x)=0} \right)^2 \right\rangle_{D_N \sim \mu_x^N} =$$

$$= \lim_{M \to 0} \lim_{\tilde{M} \to 0} \frac{1}{M\tilde{M}} \cdot \int Df_1 \dots \int Df_M \int D\tilde{f}_1 \dots \int D\tilde{f}_{\tilde{M}} \exp \left( -\frac{1}{2} \sum_{m=1}^{M} \| f_m \|_{\mathcal{H}_K}^2 - \frac{1}{2} \sum_{\tilde{m}=1}^{\tilde{M}} \left\| \tilde{f}_{\tilde{m}} \right\|_{\mathcal{H}_K}^2 \right)$$

$$\sum_{m=1}^{M} f_m(x^*) \sum_{\tilde{m}=1}^{\tilde{M}} \tilde{f}_{\tilde{m}}(x^*) \left\langle \exp \left( -\sum_{m=1}^{M} \frac{(f_m(x) - g(x))^2}{2\sigma^2} - \sum_{\tilde{m}=1}^{\tilde{M}} \frac{\left( \tilde{f}_{\tilde{m}}(x) - g(x) \right)^2}{2\sigma^2} \right)^N \right\rangle_{x \sim \mu_x}$$

Averaging w.r.t poisson distribution:

$$\left\langle \left\langle \left[f_{D_N,\sigma^2}^*(x^*)\right]^2 \right\rangle_{D_N \sim \mu_x^N} \right\rangle_{N \sim \text{Poi}(\eta)} = \sum_{N=0}^{\infty} \frac{e^{-\eta}\eta^N}{N!} \left\langle \left[f_{D_N,\sigma^2}^*(x^*)\right]^2 \right\rangle_{D_N \sim \mu_x^N} =$$

$$= \lim_{M \to 0} \lim_{\tilde{M} \to 0} \frac{1}{M\tilde{M}} \cdot \int Df_1 \ldots \int Df_M \int D\tilde{f}_1 \ldots \int D\tilde{f}_{\tilde{M}}$$

$$\exp\left(-\eta - \frac{1}{2}\sum_{m=1}^{M} \|f_m\|_{\mathcal{H}_K}^2 - \frac{1}{2}\sum_{\tilde{m}=1}^{\tilde{M}} \left\|\tilde{f}_{\tilde{m}}\right\|_{\mathcal{H}_K}^2 \right)$$

$$\exp\left(\eta \left\langle \exp\left(-\sum_{m=1}^{M} \frac{(f_m(x)-g(x))^2}{2\sigma^2} - \sum_{\tilde{m}=1}^{\tilde{M}} \frac{\left(\tilde{f}_{\tilde{m}}(x)-g(x)\right)^2}{2\sigma^2}\right)\right\rangle_{x \sim \mu_x}\right) \sum_{m=1}^{M} f_m(x^*) \sum_{\tilde{m}=1}^{\tilde{M}} \tilde{f}_{\tilde{m}}(x^*) \approx$$

$$\approx \lim_{M \to 0} \lim_{\tilde{M} \to 0} \frac{1}{M\tilde{M}} \cdot \int Df_1 \ldots \int Df_M \int D\tilde{f}_1 \ldots \int D\tilde{f}_{\tilde{M}}$$

$$\exp\left(-\frac{1}{2}\sum_{m=1}^{M} \|f_m\|_{\mathcal{H}_K}^2 - \frac{1}{2}\sum_{\tilde{m}=1}^{\tilde{M}} \left\|\tilde{f}_{\tilde{m}}\right\|_{\mathcal{H}_K}^2 + \eta \left\langle \left(-\sum_{m=1}^{M} \frac{(f_m(x)-g(x))^2}{2\sigma^2} - \sum_{\tilde{m}=1}^{\tilde{M}} \frac{\left(\tilde{f}_{\tilde{m}}(x)-g(x)\right)^2}{2\sigma^2}\right)\right\rangle_{x \sim \mu_x}\right.$$

$$\left. + \frac{\eta}{2}\left\langle \left(-\sum_{m=1}^{M} \frac{(f_m(x)-g(x))^2}{2\sigma^2} - \sum_{\tilde{m}=1}^{\tilde{M}} \frac{\left(\tilde{f}_{\tilde{m}}(x)-g(x)\right)^2}{2\sigma^2}\right)^2\right\rangle_{x \sim \mu_x}\right) \sum_{m=1}^{M} f_m(x^*) \sum_{\tilde{m}=1}^{\tilde{M}} \tilde{f}_{\tilde{m}}(x^*) =$$

$$\approx \lim_{M \to 0} \lim_{\tilde{M} \to 0} \frac{1}{M\tilde{M}} \cdot \int Df_1 \ldots \int Df_M \int D\tilde{f}_1 \ldots \int D\tilde{f}_{\tilde{M}}$$

$$\exp\left(-\frac{1}{2}\sum_{m=1}^{M} \|f_m\|_{\mathcal{H}_K}^2 - \frac{1}{2}\sum_{\tilde{m}=1}^{\tilde{M}} \left\|\tilde{f}_{\tilde{m}}\right\|_{\mathcal{H}_K}^2 + \eta \left\langle \left(-\sum_{m=1}^{M} \frac{(f_m(x)-g(x))^2}{2\sigma^2} - \sum_{\tilde{m}=1}^{\tilde{M}} \frac{\left(\tilde{f}_{\tilde{m}}(x)-g(x)\right)^2}{2\sigma^2}\right)\right\rangle_{x \sim \mu_x}\right)$$

$$\left(1 + \frac{\eta}{2}\left\langle \left(\sum_{m=1}^{M} \frac{(f_m(x)-g(x))^2}{2\sigma^2} + \sum_{\tilde{m}=1}^{\tilde{M}} \frac{\left(\tilde{f}_{\tilde{m}}(x)-g(x)\right)^2}{2\sigma^2}\right)^2\right\rangle_{x \sim \mu_x}\right) \sum_{m=1}^{M} f_m(x^*) \sum_{\tilde{m}=1}^{\tilde{M}} \tilde{f}_{\tilde{m}}(x^*) =$$

$$= \left(f_{\eta,\sigma^2}^{EK}(x^*)\right)^2$$

$$+ \lim_{M \to 0} \lim_{\tilde{M} \to 0} \frac{1}{M\tilde{M}} \cdot \frac{\eta}{8\sigma^4} \int d\mu_x \left\langle \left(\sum_{a=1}^{M} (f_a(x)-g(x))^2 + \sum_{b=1}^{\tilde{M}} \left(\tilde{f}_b(x)-g(x)\right)^2\right)^2 \sum_{c=1}^{M} f_c(x^*) \sum_{d=1}^{\tilde{M}} \tilde{f}_d(x^*)\right\rangle_0 =$$

$$= \left(f_{\eta,\sigma^2}^{EK}(x^*)\right)^2$$

$$+ \lim_{M \to 0} \lim_{\tilde{M} \to 0} \frac{1}{M\tilde{M}} \cdot \frac{\eta}{4\sigma^4} \int d\mu_x \left[\left\langle \sum_{a=1}^{M} (f_a(x)-g(x))^2 \sum_{b=1}^{M} (f_b(x)-g(x))^2 \sum_{c=1}^{M} f_c(x^*) \sum_{d=1}^{\tilde{M}} \tilde{f}_d(x^*)\right\rangle_0\right.$$

$$\left. + \left\langle \sum_{a=1}^{M} (f_a(x)-g(x))^2 \sum_{b=1}^{\tilde{M}} \left(\tilde{f}_b(x)-g(x)\right)^2 \sum_{c=1}^{M} f_c(x^*) \sum_{d=1}^{\tilde{M}} \tilde{f}_d(x^*)\right\rangle_0\right] =$$

$$= \left(f_{\eta,\sigma^2}^{EK}(x^*)\right)^2 + \frac{\eta}{4\sigma^4} \int d\mu_x \underbrace{\lim_{M \to 0} \frac{1}{M} \left\langle \sum_{a=1}^{M} (f_a(x)-g(x))^2 \sum_{b=1}^{M} (f_b(x)-g(x))^2 \sum_{c=1}^{M} f_c(x^*)\right\rangle_0}_{8\left(f_{\eta,\sigma^2}^{EK}(x)-g(x)\right)\text{Var}[f(x)]\text{Cov}[f(x),f(x^*)] \text{ as we saw in } \langle f\rangle} f_{\eta,\sigma^2}^{EK}(x^*)$$

$$+ \frac{\eta}{4\sigma^4} \int d\mu_x \left(\lim_{M \to 0} \frac{1}{M} \left\langle \sum_{a=1}^{M} (f_a(x)-g(x))^2 \sum_{b=1}^{M} f_b(x^*)\right\rangle_0\right)^2$$

and we're left with:

$$\lim_{M \to 0} \frac{1}{M} \left\langle \sum_{a=1}^{M} (f_a(x) - g(x))^2 \sum_{b=1}^{M} f_b(x^*) \right\rangle_0 =$$

$$\lim_{M \to 0} \frac{1}{M} \left[ M \left\langle (f(x) - g(x))^2 f(x^*) \right\rangle_0 + M(M-1) \langle f(x^*) \rangle \left\langle (f(x) - g(x))^2 \right\rangle_0 \right] =$$

$$= \left\langle (f(x) - g(x))^2 f(x^*) \right\rangle_0 - \langle f(x^*) \rangle_0 \left\langle (f(x) - g(x))^2 \right\rangle_0 = 2 \left( f_{\eta,\sigma^2}^{EK}(x) - g(x) \right) \mathrm{Cov}\left[ f(x), f(x^*) \right]$$

but this correction gives $O\left( \frac{1}{\eta^3} \right)$ so:

$$\left\langle \left\langle \left[ f_{D_N,\sigma^2}^*(x^*) \right]^2 \right\rangle_{D_N \sim \mu_x^N} \right\rangle_{N \sim \mathrm{Poi}(\eta)} =$$

$$\left( f_{\eta,\sigma^2}^{EK}(x^*) \right)^2 + \frac{2\eta}{\sigma^4} f_{\eta,\sigma^2}^{EK}(x^*) \int d\mu_x \left( f_{\eta,\sigma^2}^{EK}(x) - g(x) \right) \mathrm{Var}\left[ f(x) \right] \mathrm{Cov}\left[ f(x), f(x^*) \right] + O\left( \frac{1}{\eta^3} \right)$$

and notably

$$\langle f^2 \rangle = \langle f \rangle^2 + O\left( \frac{1}{\eta^3} \right)$$

## F.6 Perubative Correction for Rotationally Invariant Kernel

We now wish to evaluate this expression for a rotationally invariant kernel and a uniform measure on the hypersphere. This simplyfies the expression for $\langle f \rangle$ to:

$$f_{\eta,\sigma^2}^{GC}(x^*) =$$

$$= f_{\eta,\sigma^2}^{EK}(x^*) + \frac{\eta}{\sigma^4} \int d\mu_x \left( f_{\eta,\sigma^2}^{EK}(x) - g(x) \right) \mathrm{Var}\left[ f(x) \right] \mathrm{Cov}\left[ f(x), f(x^*) \right] + O\left( \frac{1}{\eta^3} \right) =$$

$$= f_{\eta,\sigma^2}^{EK}(x^*) + \frac{\eta}{\sigma^4} C_{K,\eta,\sigma^2} \int d\mu_x \left( f_{\eta,\sigma^2}^{EK}(x) - g(x) \right) \mathrm{Cov}\left[ f(x), f(x^*) \right] =$$

$$= f_{\eta,\sigma^2}^{EK}(x^*) - \frac{\eta}{\sigma^4} C_{K,\eta,\sigma^2} \sum_{i,j} \frac{\frac{\sigma^2}{\eta}}{\lambda_i + \frac{\sigma^2}{\eta}} \left( \frac{1}{\lambda_j} + \frac{\eta}{\sigma^2} \right)^{-1} g_i \phi_j(x^*) \underbrace{\int d\mu_x \phi_i(x) \phi_j(x)}_{\delta_{ij}} + O\left( \frac{1}{\eta^3} \right) =$$

$$= f_{\eta,\sigma^2}^{EK}(x^*) - \frac{\eta}{\sigma^4} C_{K,\eta,\sigma^2} \sum_{i} \frac{\frac{\sigma^2}{\eta}}{\lambda_i + \frac{\sigma^2}{\eta}} \left( \frac{1}{\lambda_i} + \frac{\eta}{\sigma^2} \right)^{-1} g_i \phi_i(x^*) + O\left( \frac{1}{\eta^3} \right) =$$

$$= f_{\eta,\sigma^2}^{EK}(x^*) - C_{K,\eta,\sigma^2} \sum_{l,m} \frac{g_{l.m}}{\sigma^2 \left( 2 + \frac{\lambda_l \eta}{\sigma^2} + \frac{\sigma^2}{\lambda_l \eta} \right)} Y_{l,m}(x^*) + O\left( \frac{1}{\eta^3} \right).$$

The expression for $\langle f^2 \rangle$ is:

$$\left\langle \left\langle \left[ f_{D_N,\sigma^2}^*(x^*) \right]^2 \right\rangle_{D_N \sim \mu_x^N} \right\rangle_{N \sim \mathrm{Poi}(\eta)} =$$

$$= \left( f_{\eta,\sigma^2}^{EK}(x^*) \right)^2 - 2 f_{\eta,\sigma^2}^{EK}(x^*) C_{K,\eta,\sigma^2} \sum_{l,m} \frac{g_{l.m}}{\sigma^2 \left( 2 + \frac{\lambda_l \eta}{\sigma^2} + \frac{\sigma^2}{\lambda_l \eta} \right)} Y_{l,m}(x^*) + O\left( \frac{1}{\eta^3} \right)$$

# G  VARIOUS INSIGHTS

## G.1  CORRECTION MEANS WORSE GENERALIZATION

The correction always means worse generalization than what the EK suggests. Indeed

$$
f_{\eta,\sigma^2}^{GC}\left(x^*\right) = f_{\eta,\sigma^2}^{EK}\left(x^*\right) - C_{K,\eta,\sigma^2} \sum_{l,m} \frac{g_{l,m}}{\sigma^2\left(2 + \frac{\lambda_l \eta}{\sigma^2} + \frac{\sigma^2}{\lambda_l \eta}\right)} Y_{l,m}\left(x^*\right) + O\left(\frac{1}{\eta^3}\right) =
$$

$$
= \sum_{l,m} \frac{\lambda_l}{\lambda_l + \frac{\sigma^2}{\eta}} g_{l,m} Y_{l,m}\left(x^*\right) - C_{K,\eta,\sigma^2} \sum_{l,m} \frac{g_{l,m}}{\sigma^2\left(2 + \frac{\lambda_l \eta}{\sigma^2} + \frac{\sigma^2}{\lambda_l \eta}\right)} Y_{l,m}\left(x^*\right) + O\left(\frac{1}{\eta^3}\right) =
$$

$$
= \sum_{l,m} \left( \underbrace{ \frac{\lambda_l}{\lambda_l + \frac{\sigma^2}{\eta}} - \underbrace{\frac{C_{K,\eta,\sigma^2}}{\sigma^2\left(2 + \frac{\lambda_l \eta}{\sigma^2} + \frac{\sigma^2}{\lambda_l \eta}\right)}}_{\text{positive}} }_{< \frac{\lambda_l}{\lambda_l + \frac{\sigma^2}{\eta}} < 1} \right) g_{l,m} Y_{l,m}\left(x^*\right)
$$

## G.2  EXACT EIGENVALUES FOR 2-LAYER ReLU NTK WITH $\sigma_b^2 = 0$

For the NTK associated with a 2-layer ReLU NTK without bias we were able to fined an exact expression for the eigenvalues for all $l$:

$$
\lambda_{2k} = \frac{\sigma_{w_1}^2 \sigma_{w_2}^2}{2\pi} \cdot \frac{d(1+2k) + (1-2k)^2}{8\pi} \left( \frac{\Gamma\left(k - \frac{1}{2}\right) \Gamma\left(\frac{d}{2}\right)}{\Gamma\left(k + \frac{d+1}{2}\right)} \right)^2 , \lambda_{2k+1} = \frac{\sigma_{w_1}^2 \sigma_{w_2}^2}{2\pi} \cdot \frac{\pi}{d} \delta_{k,0}
$$

It is interesting to note that for all odd $l > 1$ $\lambda_l = 0$ so the expressive power of the kernel (and hence the neural network) is greatly reduced.

# H  ACCURACY OF THE RENORMALIZED NTK

For two normalized datapoints $x$ and $x'$, drawn from a uniform dataset on a hypersphere of radius 1, and at large $d$ the random variable $(x \cdot x')$ is approximately Gaussian with variance $O(d^{-1})$. Since $(x \cdot x')$ is bounded to $[-1, 1]$, the random variable $(x \cdot x')^r$ must have a standard deviation which is decaying function of $r$. For $r \ll d$ and large $d$ one can estimate the magnitude this standard deviation from exact known expressions and a saddle-point approximation yielding $O((d/r)^{-r/2}) \approx O(d^{-r/2})$ [3]. Considering next the tail of the Taylor expansion $\sum_{q>r} b_q (x \cdot x')^q$, projected on the dataset $(\sum_{q>r} b_q (x_n \cdot x_m)^q)$. The resulting $N$ by $N$ matrix is $\sum_{q>r} b_q$ on the diagonal but $O(d^{-(r+1)/2})$ in all other entries. As we justified in the main text, our renormalization transformation amounts to keeping only the diagonal piece of this matrix and interpreting it as noise.

Consider then (1) for $g^\star$ in two scenarios: (I) $g_\infty^\star$ with the full NTK ($K(x,x')$) and no noise and (II) $g_r^\star$ with the NTK trimmed after the $r$'th power ($K_r(x,x')$) but with $\sigma_r^2 = \sum_{q>r} b_q$. The first $K(x_\star, x_n)$ piece, for $x_\star$ drawn from the dataset distribution, obeys $K(x_\star, x_n) - K_r(x_\star, x_n) = O(d^{-(r+1)/2})$. Next we compare $K_r(x_n, x_m) + I_{nm}\sigma_r^2$ and $K(x_n, x_n)$. On their diagonal they agree exactly but their off-diagonal terms agree only up to a $O(d^{-(r+1)/2})$ discrepancy. Denoting by $\delta K$ the difference between these two matrices, we may expand $K^{-1} = [K_r + \sigma_m^2 I + \delta K]^{-1} = [K_r + \sigma_r^2 I]^{-1}[1 - \delta K[K_r + \sigma_r^2 I]^{-1} + \delta K[K_r + \sigma_r^2 I]^{-1}\delta K[K_r + \sigma_r^2 I]^{-1} + ...]$.

---

[3] A more accurate estimate is $\left(\frac{r}{r+d-3}\right)^{r/2} \left(\frac{d-3}{r+d-3}\right)^{d/4}$

We next argue that $\delta K[K_r + \sigma_r^2 I]^{-1}$ multiplied by target vector $(g(x_n))$ is negligible compared to the identity for large enough $r$ thereby establishing the equivalence of the two scenarios. Indeed consider the eigenvalues of $\delta K[K_r + \sigma_r^2 I]^{-1}$. As $\delta K_{nm}$ is $O(d^{-(r+1)/2})$ its typical eigenvalues are $O(\sqrt{N} d^{-(r+1)/2})$ and bounded by $O(N d^{-(r+1)/2})$. The typical eigenvalues of $[K_r + \sigma_m^2 I]^{-1}$ are of the same order as $K(x_n, x_n) = K$ and bounded from below by $\sigma_r^2$. Thus typical eigenvalues of $\delta K[K_r + \sigma_r^2 I]^{-1}$ are $O(\sqrt{N} d^{-(r+1)/2}/K)$ and bounded from above by $O(N d^{-(r+1)/2}/\sigma_r^2)$. The NTK has the desirable property that $\sigma_r^2$ decays very slowly. Thus certainly in the typical case but even in the worse case scenario we expect good agreement at large $r$. In Fig. 1, right panel, we provide supporting numerical evidence.

We refer to $K_r(x, x')$ as the renormalized NTKs at the scale $r$. As follows from (13), $\lambda_l$'s with $l \geq r$ are zero. Therefore, as advertised, the high-energy-sector has been removed and compensated by noise on the target and a change of the remaining $l < r$ (low-energy) eigenvalues. A proper choice of $r$ involve two considerations. Requiring perturbation theory to hold well ($C_{K_r, \sigma_r^2/\eta} < \sigma_r^2$) which puts an $\eta$-depended upper bound on $r$ and requiring small discrepancy in predictions puts another $\eta$ dependent lower bound on $r$ (typically $\sqrt{N} d^{-(r+1)/2} \ll 1$).

Lastly we comment that our renormalization NTK approach is not limited to uniform datasets. The entire logic relies on having a rapidly decaying ratio of off-diagonal moments $((x_n \cdot x_m)^{2r})$ and diagonal moments $(x_n \cdot x_n)^{2r}$ as one increases $r$. We expect this to hold in real-world distributions. For instance for a multi-dimension Gaussian data distribution the input dimension $(d)$ traded by an effective dimension $(d_{eff})$ defined by the variance of $(x_m \cdot x_n)$.

