# OpenReview forum: "Learning Curves for Deep Neural Networks: A field theory perspective"
_ICLR.cc/2020/Conference — Reject_

### Official Review · AnonReviewer2 · 2019-10-23
**Official Blind Review #2**

**Rating:** 3

**Review:**

This paper explores how tools from perturbative field theory can be used to shed light on properties of the generalization error of Gaussian process/kernel regression, particularly on how the error depends on the number of samples N. For uniform data on the sphere, a controlled expansion is obtained in terms of the eigendecomposition of the kernel. Although the expansion breaks when the noise term goes to zero, a renormalized kernel is introduced for which an accurate perturbative expansion is possible. A variety of empirical results confirm the theoretical analysis.

The results presented here are interesting and I particularly liked the introduction of the renormalized kernel to study the noiseless case. The agreement in Fig 1 is quite impressive, and the improvements relative to naive 1/sqrt(N) scaling as highlighted in App. C shows the power of the approach. The topic is salient and will interest most theoretically-minded researchers, and I think there is an abundance of new ideas and novel content. My only real concern is with the presentation.

This is a fairly technical paper that utilizes a substantial amount of physics jargon and many loose, hand-wavy arguments that rely on significant amount of prior field theory knowledge. I suspect that only a small fraction of the community will have the adequate background to get much out of this paper. For publication in a machine learning conference, I think more effort should be devoted to speaking to the machine learning audience. Some ways to achieve this might include reorganizing the technical points into bite-size chunks, laying out a roadmap for the main calculations and results, highlighting the important takeaways, including more figures, and concretely emphasizing the connections to practice and prior work.

Overall, I am a bit on the fence, but leaning towards rejection for the above reasons. I could be convinced to increase my score if I am reassured that non-physicists are able to follow the arguments and find this paper interesting.

**Experience Assessment:**

I have published in this field for several years.

**Review Assessment: Checking Correctness Of Derivations And Theory:**

I assessed the sensibility of the derivations and theory.

**Review Assessment: Checking Correctness Of Experiments:**

I assessed the sensibility of the experiments.

**Review Assessment: Thoroughness In Paper Reading:**

I read the paper at least twice and used my best judgement in assessing the paper.

---

> ### Author Response · Authors · 2019-11-13
> **Review Response**
>
> We appreciate the time spend by the referee on reviewing our work. We are happy to he or she found it interesting. Indeed the presentation of the previous version has been less tailored for a machine learning audience. Accordingly we have made major changes to the presentation to make it more widely appealing. Specifically: We emphasize the results over the tools whenever appropriate and clarify all definitions. We also compartmentalized the field-theory section to a particular  sub-section which can be skipped without comprising the message and tried show the derivation road map in smaller increments.  We also included a clearer explanation of the main results, the experiment, and the intuition regarding the renormalized NTK.
>
> Following these changes, the basic results of our work should stand-out more clearly. We re-state them now to stress their wider-ML-audience appeal
>
> 1. On uniform datasets, deep fully connected networks are not a black-box. One can understand what they do very accurately with pen and paper.
> 2. Noiseless GP inference can be mapped onto noisy GP inference by trimming the Taylor expansion, in what can be thought-of as a form of renormalization where one coarse grains the angular resolution. The NTK kernels are effective in creating a lot of noise following this normalization.
> 3. As now explained more clearly in Sec. 6, deep fully connected networks trained in the NTK regime (with infinite number of parameters) do not suffer from over fitting due to an implicit bias to low order polynomials.
> 4. Field theory tools can provide an accurate formalism to analyze DNNs.
>
> We believe this work and other physics-style computations submitted to ML conference, can safely find their place more specialized physics journals. However we feel that it is important to try and prevent an "eco chamber" of physicists talking about ML and an ML community disconnected from physics tools and methodologies. Given the other reviews, the referee is in a unique position to make this call.

---

### Official Review · AnonReviewer3 · 2019-10-24
**Official Blind Review #3**

**Rating:** 8

**Review:**


This theoretical paper exploits a recent rigorous correspondence between very wide DNNs (trained in a certain way) and Neural Tangent Kernel (a case of Gaussian Process-based noiseless Bayesian Inference).
A field-theory formalism was developed for Gaussian Processes (2001). Here it is thus extended to the NTK case. There are 3 important theoretical results which are both proven and backed by numerical confirmation.  These results, in particular the first, provide a very accurate prediction for the learning curve of some models.  The paper is well situated within this literature.  I am not very knowledgeable about NTK or even Gps, however I understand the challenges of understanding DNNs and I am familiar with field theory and renormalization group.
Given the importance and quality of the results, and the overall quality and clarity of this (dense) paper, I recommend acceptation without hestiation.

There are a couple of points however that could be improved, that would make the paper more useful for the community.
Given the density of results in the paper, I would relax the length constraint, allowing up to 9 or 10 pages if possible, to add more explanations (not computations).


I would like the paper to present more explicitly how the regression target labels g(x) are generated. Maybe it is said but I couldn’t easily understand, for sure, how they are generated.

Also, please explain early enough what is meant by uniform dataset (I understood it simply means the data x is drawn uniformly at random over a manifold, here this manifold is often the d-dimensional hypersphere).

Claim II states that ‘...lead to clear relations between deep fully-connected networks and polynomial
regression’’. This is, I believe, supported by theoretical proof and numerical double-check, however it is not discussed enough for the abstract’s promise to be fulflled ‘a coherent picture emerges wherein fully-connected DNNs ...’.
I think this point deserves a more ample discussion in section 7.

More generally, the claims in the introduction or at the end of section 3 are stated rather explicitly, but very densely, and the careful reader can get the hypothesis of each result from the text.
However for the sake of ease of read of less patient readers, I think it would be appropriate to have, somewhere, a more self-contained description of the results’ list. This paper is technical and some readers will be interested of simply knowing the hypothesis made and type of results obtained.
For instance, the sentence ‘‘They [results] hold without any limitations on the dataset or the kernel and yield a variant of the EK result along with its sub-leading correction.’’ is misleading: as stated in the previous sentence in the text, this is for the fixed-teacher learning curve, etc.

Please try to explain a bit more the intuition behind renormalization / trimming terms q>r (r integer fixed, the higher the less approximated).  More specifically, it is not very clear to me how it can be interpreted in terms of how we look at the data. You mention (x.x’)^r being negligible or not depending on r,d etc, but I wounder if there is some kind of simple ‘geometrical’ interpretation (is it a coarse graining of the data in angular space, the ‘high energy’ eigenvalues corresponding to the high frequency, high resolution distinction between very close angles ?).  On that point I am a bit lost and it’s a pity because your results are strong and rely on few, rather simple/elegant assumptions (which call for some intuitive understanding).

Could you explain intuitively, to the inexperienced reader, why the noiseless case is harder to deal with than the finite-noise one ?

In appendix D, it is mentioned that you need averaging over about 10 samples to have a decent average. For a single realization of a N-sized training set, there is additional variation (Adding or subtracting to the error epsilon).  Given actual experiments are typically performed for a single realization of the data, I think this point should be mentioned in the main text more explicitly. Ideally, you could add error bars to the data, accounting for the dispersion inherent to a single-realization case.

In early section 3, a short definition of a GP should be provided (there, or before).

Around Eq. 2, you should specify the interpretation of P_0[f]

After Eq. 3, ‘where F is some functional of f.’ I would add: ‘[where F is some functional of f,] for instance Eq. 2.’
The derivation of eq 6 is not obvious. You do detail it in the appendix, but forgot to cite the appendix !

‘Notably none of these cited results apply in any straightforward manner in the NTK-regime.’ : could you quickly explain why (no matched priors ? Noise ?)

‘‘The d^-l scaling of eigenvalues’’ : at this point, the variable ‘l’ had not been defined.

‘‘notably cross-talk between features has been eliminated’’ : has it been eliminated or does it simply become constant ?

‘3% accuracy’ [relating to figure 1]
I understand the idea but accuracy seems misleading. I would replace everywhere with something like ‘relative mismatch’.  OR explain better why you call this accuracy: usually a high accuracy is preferred, and here you are proud with this very low imprecision of 3%

‘’Taking the leading order term one obtains the aforementioned EK result with N’’ : maybe (just a suggestion here) you could recall it here, given it was in page 1 (and in-line).


Appendix B:  could you explain why this difference increases with N ? I would have expected this kind of quantity to decrease with N.

Appendix F: there are typos in the r.h.s. in the first line.
\sum_j f_j \phi_j  (I think).

Appendix G: ‘noisy targets’ : you mean fully random or Kernel + some degree of noise with variance sigma^2 ?  I think it’s the first time you use this phrasing.

Appendix G: you denote \partial / \partial \alpha for the functional derivative. I would replace with \delta to stress out it is a functional and not regular derivative.

Beyond	 appendix G.1 : I confess I didn’t have time to read it.

Despite the overall quality of the text, there are a number of wrong singular/plural matchings, which can easily be corrected. Here are some of them, with other typos as well:
‘Furthermore since our aim was to predict what the DNNs would predict rather [THAN?] reach SOTA predictions’

‘a factor of a factor
of about 3.’

as do for – > as we do for

uniformally - > uniformly

**Experience Assessment:**

I have read many papers in this area.

**Review Assessment: Checking Correctness Of Derivations And Theory:**

I assessed the sensibility of the derivations and theory.

**Review Assessment: Checking Correctness Of Experiments:**

I carefully checked the experiments.

**Review Assessment: Thoroughness In Paper Reading:**

I read the paper thoroughly.

---

> ### Author Response · Authors · 2019-11-13
> **Review Response Part I**
>
> We appreciate the time spent by the referee on reviewing our work. We are happy to he or she found it interesting. The referee, along with the other referees, made several important comments regarding presentation which we took very seriously. We believe the revised manuscript delivers the message much more effectively and for a wider machine learning audience. Specifically: We emphasize the results over the tools whenever appropriate and clarify all definitions. We also compartmentalized the field-theory section to a particular sub-section which can be skipped without comprising the overall message. We also included clearer explanation of the experiment and extended the manuscript by one page, to 9 pages.
>
> We next address the referee's specific question/comments below.
>
> R. I would like the paper to present more explicitly how the regression target labels g(x) are generated. Maybe it is said but I couldn't easily understand, for sure, how they are generated.
>
> A. A far more detail explanation of the experiment now appears in Sec. 6.
>
> R. Also, please explain early enough what is meant by uniform dataset (I understood it simply means the data x is drawn uniformly at random over a manifold, here this manifold is often the d-dimensional hypersphere).
>
> A. This is correct and now appears in the introduction.
>
> R. Claim II states that ‘...lead to clear relations between deep fully-connected networks and polynomial
> regression’’. This is, I believe, supported by theoretical proof and numerical double-check, however it is not discussed enough for the abstract’s promise to be fulfilled ‘a coherent picture emerges wherein fully-connected DNNs ...’.
> I think this point deserves a more ample discussion in section 7.
>
> A. Section 6. includes an additional paragraph which explains this point by combining the ideas of previous section and the bound we obtain on the eigenvalues of any renormalized NTK.
>
> R. More generally, the claims in the introduction or at the end of section 3 are stated rather explicitly, but very densely, and the careful reader can get the hypothesis of each result from the text.
> However for the sake of ease of read of less patient readers, I think it would be appropriate to have, somewhere, a more self-contained description of the results’ list. This paper is technical and some readers will be interested of simply knowing the hypothesis made and type of results obtained.
> For instance, the sentence ‘‘They [results] hold without any limitations on the dataset or the kernel and yield a variant of the EK result along with its sub-leading correction.’’ is misleading: as stated in the previous sentence in the text, this is for the fixed-teacher learning curve, etc.
>
> A. We have improved our results list, disentangled the derivations from the results (see for instance sub-section 3.3). Regarding fix-teacher versus average, we agree with the referee on the technical point however note that fix-teacher learning curves are typically harder to obtain compared to ones derived from say a different Gaussian prior. More specifically it is very easy now to take our expressions for the MSE error and average them over any teacher prior for which $\langle g_n g_m \rangle_{Teachers}$ in known. This is because $g$ appears linearly in all of our predictions.   R. Please try to explain a bit more the intuition behind renormalization / trimming terms q>r (r integer fixed, the higher the less approximated).  More specifically, it is not very clear to me how it can be interpreted in terms of how we look at the data. You mention $(x\cdot x’)^r$ being negligible or not depending on r,d etc, but I wounder if there is some kind of simple ‘geometrical’ interpretation (is it a coarse graining of the data in angular space, the ‘high energy’ eigenvalues corresponding to the high frequency, high resolution distinction between very close angles ?).  On that point I am a bit lost and it’s a pity because your results are strong and rely on few, rather simple/elegant assumptions (which call for some intuitive understanding).
>
> A. We perfectly agree with referee's geometric interpretation. Section 5. now contains a paragraph discussing the intuition behind our renormalizaiton group approach.
>
> R. Could you explain intuitively, to the inexperienced reader, why the noiseless case is harder to deal with than the finite-noise one ?
>
> A. Such an intuition now appears in the text, below Eq. (4). The relies on the notion that hard constraints are typically less tractable than soft constraints and similarly that averaging makes problems easier as it reduces the amount of information.

---

> > ### Author Response · Authors · 2019-11-13
> > **Review Response Part II**
> >
> > R. In appendix D, it is mentioned that you need averaging over about 10 samples to have a decent average. For a single realization of a N-sized training set, there is additional variation (Adding or subtracting to the error epsilon).  Given actual experiments are typically performed for a single realization of the data, I think this point should be mentioned in the main text more explicitly. Ideally, you could add error bars to the data, accounting for the dispersion inherent to a single-realization case.
> >
> > A. The error bars would be of the order of 0.001\% and completely invisible in our graph. The reason is that while we only take 10~ samples per dataset size (and consequently the dataset averaged generalization error does have a noticeable 5\% relative error), we also perform the Poisson averaging over many datasets of similar size. This greatly suppresses the errors. We now address this point explicitly in Sec 6 (in addition to demonstrating it in appendix A).
> >
> > R. In early section 3, a short definition of a GP should be provided (there, or before).
> >
> > A. Done. See Sec. 3.1
> >
> > R. Around Eq. 2, you should specify the interpretation of $P_0[f]$
> >
> > A. Done. See Sec. 3.2
> >
> > R. After Eq. 3, ‘where F is some functional of f.’ I would add: ‘[where F is some functional of f,] for instance Eq. 2.’
> > The derivation of eq 6 is not obvious. You do detail it in the appendix, but forgot to cite the appendix !
> >
> > A. Done. See Sec. 3.2.
> >
> > R. ‘Notably none of these cited results apply in any straightforward manner in the NTK-regime.’ : could you quickly explain why (no matched priors ? Noise ?)
> >
> > A. When preparing the original manuscript we tediously went through all these cited results as explained in the prior works section. The main issue is that very few treat the noiseless case beyond one dimension and two dimensional settings. The one work we know which makes predictions at higher dimension (Sollich (2001)) works in the teacher-averaged predictions (rather than fixed teacher), makes assumptions on some level of matching between the teacher-prior and the GP-prior, and doesn't lead to explicit expression.
> >
> > R. ‘‘The $d^{-l}$ scaling of eigenvalues’’ : at this point, the variable ‘l’ had not been defined.
> >
> > A. Fixed.
> >
> > R. ‘‘notably cross-talk between features has been eliminated’’ : has it been eliminated or does it simply become constant ?
> >
> > A. It has been eliminated since only $g_i$ affects the coefficient of $\phi_i(x)$ in the predictions. Learning becomes diagonal in feature space.
> >
> > R. ‘3% accuracy’ [relating to figure 1]
> > I understand the idea but accuracy seems misleading. I would replace everywhere with something like ‘relative mismatch’.  OR explain better why you call this accuracy: usually a high accuracy is preferred, and here you are proud with this very low imprecision of 3%
> >
> > A. We thank for the referee for this comment. This has been changed throughout the text.
> >
> > R. ‘’Taking the leading order term one obtains the aforementioned EK result with N’’ : maybe (just a suggestion here) you could recall it here, given it was in page 1 (and in-line).
> >
> > A. Aiming to make it less dense, the new introduction doesn't include the explicit expression for the EK result.
> >
> > R. Appendix B:  could you explain why this difference increases with N ? I would have expected this kind of quantity to decrease with N.
> >
> > A. As is explained more clearly in Sec. 5. While the off-diagonal terms are small there are more and more of them as the the dataset size increases (and hence the size of the matrix K(D)). Collecting all these omitted contributions to the prediction, they would appear, roughly, as a random matrix multiplying the target vector in Eq. (1). Therefore they would sum together (incoherently/with-alternating-signs) and give an error which increases with $N$. A different view point is that the large one takes $N$ the finer the features ones can resolve, and hence neglecting the high angular momentum spherical Harmonics becomes less and less adequate.
> >
> > R. All later comments (typos and related mistakes)
> >
> > A. We thank the referee for paying such detailed attention to our appendices. Most of these points have been addressed in the revised version.

---

### Official Review · AnonReviewer4 · 2019-10-29
**Official Blind Review #4**

**Rating:** 1

**Review:**

This paper used the field-theory formalism to derive two approximation formulas to the expected generalization error of kernel methods with n samples. Experiments showed that the sub-leading approximation formula approximates the generalization error well when $n$ is large.


This paper is poorly written. Many mathematic notations and terminologies are not well-defined. The setup of the experiments are not given clearly. Here I gave some examples:
1) The authors claimed that they derived two approximation formulas, EK and SL. I didn't find a clear statement in the main text saying which formula is the EK approximation and which formula is the SL approximation. My conjecture is that, Eq. (10) gives the EK formula, and SL formula is not given in the main text. In addition, Eq. (10) is confusing because the authors wrote that Eq. (10) is a simplification of Eq. (8). However, Eq. (8) was an approximate equality, then Eq. (10) turned into an equality.
2) Figure 1 gives experimental results. However, the description of the experimental setup is completely vague. The authors described the kernel and the target function as "the NTK kernel implied by a fully connected network of depth 4 with $σ_w^2 = σ_b^2 = 1$ and ReLU activations" and "a target function with equal spectral weights at $l = 1, 2$", without other explanations. I don't think readers can figure out what is exactly the kernel and the target function from this description.
3) I am concerned about the writing style of this paper. I am OK with the physics jargon the authors used in the paper, as well as the non-rigorous of the result. But I think the authors should write equations in a clear way. For example, the definition of renormalized NTK should better be defined in equations such as $K_r(x, x') = \sum_{k = 0}^r b_k <x, x'>^k$, rather than be described in words like "trim after the r’th power".


Here is a technical question:
- It seems that the authors claimed that both EK and SL give approximation error O(1/N^3). Then why SL is the "sub-leading asymptotics"?


I feel the content of the paper is somewhat interesting. However, the paper is poorly written. The authors failed to deliver effective scientific communication to the readers. The results cannot be reproduced after reading this paper. Therefore, I would give a clear reject.


---------
After reading the response and the revised paper:

I found that the authors modified and improved their manuscript a lot. They made much effort to address the issues I raised. This is why I think I can potentially raise my score to a weak rejection.

However, the modifications made by the authors are still not sufficient. For example, I asked the authors in my review to clarify what is the target function for the experiments. The authors now write in the paper "We consider input data in dimension d = 50 and a scalar target function $g(x) = \sum_{l=1,2;m} g_{lm}Y_{lm}(x)$ such that $\sum_{l=1, m} g_{lm}^2 = \sum_{l=2, m} g_{lm}^2 = 1/2$, but otherwise iid $g_{lm}$’s." I believe that a (random) target function that satisfies all these conditions doesn't exists. I guess what the authors want to say is something like "taking $g(x) = \sum_{l=1, 2} \sum_{m = 1}^{M_l} g_{lm}Y_{lm}(x)$, $(g_{11}, ..., g_{1 M_1}) \sim Unif(S^{M_1 - 1}(1/\sqrt 2))$, and $(g_{21}, ..., g_{2 M_2}) \sim Unif(S^{M_2 - 1}(1/\sqrt 2))$". The problem of the statement of the authors is that, if $\sum_{m=1}^{M_1} g_{1m}^2 =\sum_{m = 1}^{M_2} g_{2m}^2 = 1/2$, $(g_{lm})_{l = 1, 2; m \in \{1, \ldots, M_l \}}$ cannot be i.i.d. (one choice is to make $g_{11} = ... = g_{1M_1} = \sqrt{1/(2 M_1)}$ and $g_{21} = ... = g_{2 M_2} = \sqrt{1/(2 M_2)}$ be deterministic, but they are unequal). This is just an example of the writing problem of the paper. There are many other issues.

I doubt this paper can be easily accepted at a Physics venue. I used physics tools like replica methods and I knew some Physicists published in machine learning conferences. The papers these Physicists wrote deliver clear scientific communications, though also using jargons and non-rigorous tools. There are many papers using physics tools studying machine learning problems, which published at ML conferences like ICLR, ICML, and NeurIPS. This paper is far less as accessible as those papers.

Finally, I want to point out that, the generalization of kernel methods have been intensively studied in the machine learning literature, for example using the RKHS theory. It would be nice to cite related literature and compare the results. It is my fault that I didn't bring this point up in my review.

I agree that there could potentially be great ideas in this paper. The conference is a venue with quality control. I encourage the authors to submit this paper again after they make more efforts to improve its accessibility.

**Experience Assessment:**

I have published one or two papers in this area.

**Review Assessment: Checking Correctness Of Derivations And Theory:**

I assessed the sensibility of the derivations and theory.

**Review Assessment: Checking Correctness Of Experiments:**

I assessed the sensibility of the experiments.

**Review Assessment: Thoroughness In Paper Reading:**

I read the paper at least twice and used my best judgement in assessing the paper.

---

> ### Author Response · Authors · 2019-11-13
> **Review Response**
>
> We appreciate the time spent by the referee on reviewing our work. It is unfortunate that we couldn't communicate it better. However the reviewer may get a sense from the other referees' views, that there is a unique result "hidden" in this text. This result required much analytical effort and several novel ideas (such as the renormalized NTK). Indeed it is often said that deep neural networks are a black-box whereas, in this fairly complicated setting, we understand almost exactly what they are doing using only pen and paper. The bigger promise here is that field-theory can deliver a concrete, detailed, and accurate formalism for reasoning about DNNs. The notion of adding noise to a GP by trimming its Taylor expansion may also resonate with a wider audience.
>
> Regarding presentation, we concede that the delivery of many parts was sub-optimal and emphasized techniques over results. In addition more attention should have been placed on making the definitions clearer. We have therefore made various changes to the presentation: We emphasize the results over the tools whenever appropriate and clarify all definitions. We also compartmentalized the field-theory section to a particular sub-section which can be skipped without compromising the overall message.
> A more persistent hurdle in effectively communicating our results, is that we use physics tools and methodologies: We make reasonable assumptions which lead to experimentally verifiable/falsifiable theoretical predictions which we then proceed to test. We also use tools like field-theory, which have no axiomatic basis, because past experience in particle physics convinced us that they perform well and similarly important - that they are insightful. Although this puts strains on a reader coming from a different background, we believe it would at the long run, benefit the machine learning community as a whole. We hope that the referee accepts this as reasonable.
>
> Assuming we haven't used up all the referee's patience, we hope she or he would be willing to re-review the revised version.
>
> Regarding the referee's specific question
>
> "It seems that the authors claimed that both EK and SL give approximation error O($1/N^3$). Then why SL is the "sub-leading asymptotics".
>
> The expression we refer to as EK and SL have now been clearly defined in the text. To focus the discussion let's assume a target function which has a finite number of non-zero $g_n$'s. One can see that the ($g^{\star}_{EK,\eta}-g$) (the error in the EK results) has a leading power of $O(1/N)$, whereas the $SL$ term has a leading power of $O(1/N^2)$.

---

### Decision · Program_Chairs · 2019-12-19

**Decision:**

Reject

**Comment:**

This paper studies deep neural network (DNN) learning curves by leveraging recent connections of (wide) DNNs to kernel methods such as  Gaussian processes.

The bulk of the arguments contained in this paper are, thus, for the "kernel regime" rather than "the problem of non-linearity in DNNs", as one reviewer puts it.
When it comes to scoring this paper, it has been controversial. However a lot of discussion has taken place. On the positive side, it seems that there is a lot of novel perspectives included in this paper. On the other hand, even after the revision, it seems that this paper is still very difficult to follow for non-physicists.

Overall, it would be beneficial to perform a more careful revision of the paper such that it can be better appreciated by the targeted scientific community.